# Transfer learning enables identification of multiple types of RNA modifications using nanopore direct RNA sequencing

You Wu [1], Wenna Shao[1], Mengxiao Yan[2], Yuqin Wang[2], Pengfei Xu[1], Guoqiang Huang[1], Xiaofei Li[1], Brian D. Gregory [3], Jun Yang [2,4] ✉, Hongxia Wang [2,4] ✉ & Xiang Yu [1] ✉

Nanopore direct RNA sequencing (DRS) has emerged as a powerful tool for RNA modification identification. However, concurrently detecting multiple types of modifications in a single DRS sample remains a challenge. Here, we develop TandemMod, a transferable deep learning framework capable of detecting multiple types of RNA modifications in single DRS data. To train high-performance TandemMod models, we generate in vitro epitranscriptome datasets from cDNA libraries, containing thousands of transcripts labeled with various types of RNA modifications. We validate the performance of TandemMod on both in vitro transcripts and in vivo human cell lines, confirming its high accuracy for profiling m6A and m5C modification sites. Furthermore, we perform transfer learning for identifying other modifications such as m7G, Ψ, and inosine, significantly reducing training data size and running time without compromising performance. Finally, we apply TandemMod to identify 3 types of RNA modifications in rice grown in different environments, demonstrating its applicability across species and conditions. In summary, we provide a resource with ground-truth labels that can serve as benchmark datasets for nanopore-based modification identification methods, and TandemMod for identifying diverse RNA modifications using a single DRS sample.

Eukaryotic messenger RNAs (mRNAs) possess multiple types of modifications such as N6-methyladenosine (m6A)[1,2], 5-methylcytosine (m5C)[3], 5-hydroxymethylcytosine (hm5C)[4], pseudouridine (Ψ)[5,6], and inosine (I), which is produced from A-to-I editing[7]. Recent findings have revealed that these modifications are essential for the normal growth and development of eukaryotes[8–13]. For instance, m6A depositions destabilize specific mRNAs in embryonic stem cells, and aberrant m6A modifications are associated with various human diseases[14–16]. In plants, hyperactivation of a transgenic human m6A demethylation enzyme FTO in rice and potato increases crop yield and biomass, while decreasing m5C modification attenuates rice tolerance to high temperature[17,18]. Thus, research focused on mRNA modifications is an important area of inquiry.

Cutting-edge technologies have been developed to illustrate the complex landscape of the eukaryotic epitranscriptome[19–24]. Methods such as MeRIP-Seq[25], m6ACE-Seq[26], miCLIP[27], GLORI[2], Pseudo-seq[28] and BisSeq[29], which utilize either antibodies or chemical treatments followed by next-generation sequencing (NGS), have gained extensive use for profiling mRNA modifications. Moreover, RBS-seq has shown its capacity to simultaneously detect m5C, Ψ, and m1A transcriptome-

[1]Joint International Research Laboratory of Metabolic & Developmental Sciences, School of Life Sciences and Biotechnology, Shanghai Jiao Tong University, Shanghai 200240, China. [2]Shanghai Key Laboratory of Plant Functional Genomics and Resources, Shanghai Chenshan Botanical Garden, Shanghai 201602, China. [3]Department of Biology, University of Pennsylvania, Philadelphia, PA 19104, USA. [4]Chenshan Scientific Research Center of CAS Center for Excellence in Molecular Plant Sciences, Shanghai 201602, China. ✉e-mail: jyang03@cemps.ac.cn; hxwang@cemps.ac.cn; yuxiang2021@sjtu.edu.cn

wide based on NGS[30]. Combining with computational methods, Direct RNA Sequencing (DRS), developed by Oxford Nanopore Technology (ONT)[31], has revolutionized the field by enabling the identification of individual mRNA[32–45] or tRNA[46,47] modifications at single nucleotide resolution. During nanopore sequencing, modified bases disrupt the expected current signal when passing through the nanopores, allowing for the identification of modifications using machine learning approaches. Current DRS-based methods can be mainly categorized into comparative and de novo prediction models. Comparative methods such as DRUMMER[48] and xPore[39] have demonstrated good performance but require negative control samples[49], limiting their application scope. De novo prediction models, such as nanom6A[50], m6Anet[51], DENA[42], Penguin[41], are trained on labeled datasets from either in vitro synthetic sequences or in vivo transcribed mRNAs. Training sets derived from in vitro synthetic sequences, such as Curlcake[52] and ELIGOS[38], provided ground-truth labels for modifications. Using these datasets, several models for modification detection have been developed, successfully achieving single-base resolution modification identification at single transcript level[36,50]. However, in vitro synthetic sequences have limited diversity in sequence context, and thus the accuracy of models trained on synthetic sequences may decease when predicting naturally occurring sequences. Training datasets obtained from in vivo DRS data exhibit greater sequence complexity, however, the modification labels deduced from antibody-based experiments, such as m6A-seq[26] and miCLIP[27], are ambiguous given each site has a dynamic modification rate detected by mRNA DRS reads.

A recent study utilized nanopore DNA sequencing to identify multiple types of DNA methylations[53], but uncovering the epitranscriptome landscape with multiple types of RNA modifications using DRS remains a challenge[54]. To address this gap, we generated in vitro epitranscriptome (IVET) datasets from plant cDNA libraries producing thousands of mRNA transcripts. While numerous RNA modifications exist in eukaryotic mRNAs, the majority of them are present at low levels. Notably, m6A, m5C, and m1A are three of the most prevalent modifications[55,56]. Therefore, in this study, we generated transcripts labeled with m6A, m1A and m5C modifications, and developed a DRS-based Transferable deep learning model of multiple Modification (TandemMod) with these IVET datasets. The TandemMod models can identify multiple types of RNA modifications (m6A, m1A, m5C, hm5C, m7G, I and Ψ) in single eukaryotic DRS data at single-base resolution. We validated the TandemMod models using both independent synthetic RNA transcripts in vitro with ground-truth labels and human mRNA transcripts in vivo with labels identified by Illumina-based methods. Finally, we profiled the epitranscriptomic map of rice RNA that encompasses multiple types of modifications. Taken together, our study presented an approach for creating cDNA-library based IVET datasets, which can serve as benchmark datasets for DRS-based machine learning methods, and a deep learning model, TandemMod, for discovering multiple types of RNA modifications in single DRS data.

## Results

### Alterations in DRS features at base level and current level induced by RNA modifications

In the field of nanopore sequencing, the presence of modifications will cause fluctuations in the current signal, which typically leads to a decrease in basecalling qualities and an increase in basecalling errors[52,57]. Previous studies have shown that basecalling errors can be utilized for identifying m6A[37,52] and Ψ[36]. In particular, the mean, median, standard deviation and width of signals from 5-mer motifs have been used as features by nanom6A[50] to identify m6A sites. First, using in vitro-transcribed datasets generated in the Nookaew lab (denoted as ELIGOS datasets)[38], we calculated these 5 base-level features (mean, median, standard deviation, length of signals and per-base quality) for 6 types of modified bases (m1A, m6A, m5C, hm5C, m7G and Ψ) using

several motifs as examples and compared them to those that are unmodified. Our results showed that all of these modification types caused variations in base-level values and these variations were modification-specific (Fig. 1a). For instance, the mean and median signal of m5C, hm5C, m6A, m1A, and m7G were significantly increased compared to corresponding unmodified bases in the given 5-mer motif. m7G led to a decrease in standard deviation, while Ψ caused an increase in standard deviation. The signal length (dwell) of hm5C and m6A was significantly longer, while the dwell of m7G was much shorter. Additionally, we found that the base quality of hm5C, m5C, m1A and Ψ was significantly decreased (Fig. 1a). Consistently, the per-read quality of modified samples dramatically decreased compared to the unmodified sample (Fig. 1b). Notably, the variation patterns of the base features such as mean depended on sequence context in different 5-mer motifs (Supplementary Fig. 1a). Therefore, the 5-mer sequences were also considered as an additional base-level feature.

Next, we investigated the impact of modifications on nanopore current fluctuations. In nanopore sequencing, the electric current signal level data produced from a nanopore read is referred to as a squiggle. After basecalling, the raw reads may contain some errors compared to the reference sequences. Therefore, squiggling is needed to define a new assignment from squiggles to the reference sequences (For more details, refer to Methods). After squiggling, we obtained current signals corresponding to each nucleotide. However, the varying length of current signals per base poses a challenge for extracting features at the current level. To address this issue, we performed signal resampling with spline interpolation to obtain signals of equal length (100 time points per base in this study) (Supplementary Fig. 1b). The resampled signals displayed a strong positive correlation with raw signals (Supplementary Fig. 1c), and therefore can effectively represent modification features. Current intensity alterations were observed in modified bases and their neighboring bases (Fig. 1c). The resampled signals corresponding to each 5-mer sequence exist in a 500-dimensional space. We applied UMAP[58] transformation, a manifold learning technique utilized for dimensionality reduction, to the resampled signals, converting them into 2-dimensional data. The results showed that the representative 5-mer sequences with modified and unmodified bases tended to be distributed in different regions (Fig. 1d). Subsequently, we explored whether modified bases could be distinguished from the other 3 canonical bases at the current level. Our results demonstrated that the difference between modified and unmodified bases was smaller than the difference observed among the other 3 canonical bases (Supplementary Fig. 1d, e). This indicated that current intensity can be effectively utilized for modification identification.

Additionally, we also explored the base-level and current-level features of 5moU and I in ELIGOS datasets (Supplementary Fig. 2a–d). For the CAUCA motif, the mean and median signal, as well as the base quality, between U and 5moU were significantly different, while for the AUGUU motif, the standard deviation (std), signal length (dwell) and base quality between G and I were remarkably changed. Collectively, our analysis of various modifications, including m1A, m6A, m5C, hm5C, m7G, 5moU, Ψ and I, revealed that all of these modifications were associated with notable alterations in both current-level and base-level features. This highlighted the potential utility of these features for accurate identification of diverse modifications.

### TandemMod: a deep learning frame developed to detect RNA modifications at single base resolution

After conducting a systematic analysis of individual feature alterations in DRS data caused by modified bases, we extracted current intensity with 100 time points, and 6 base-level statistical characteristics (base type, base quality, mean, std, median and dwell) as features for each base. We anticipated that the use of multiple features would enhance the sensitivity and specificity of modification identification, allowing

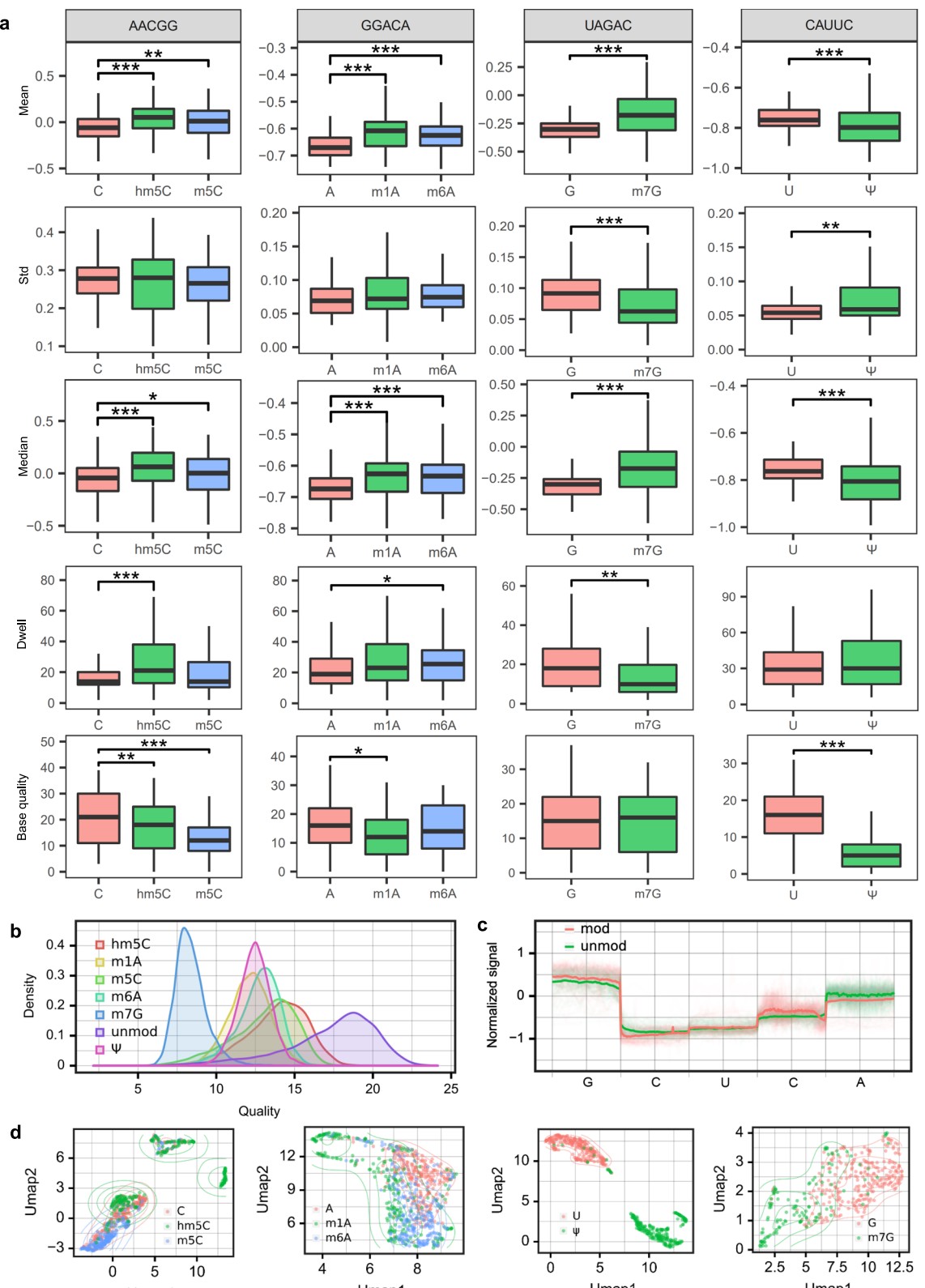

**Fig. 1 | Systematic analysis of Alterations of DRS features at base level and current level contributed by RNA modification. a** Base-level features extracted from ELIGOS datasets, which included m¹A, m⁶A, m⁵C, hm⁵C, m⁷G, and Ψ. The boxplot showing the mean, standard deviation (std), median, width of signals (dwell) and base quality of each modified base and modification-free bases, which are in the center of 4 representative 5-mer motifs. The upper and lower limits represent the 75th and 25th percentiles, respectively, while the center line represents the median; upper and lower whiskers indicate ±1.5× the interquartile range.

Outliers are not shown in these figures. All statistical tests used two-sided Wilcoxon tests. Significance levels are: \*$p < 0.05$, \*\*$p < 0.01$, \*\*\*$p < 0.001$. The exact $p$ values are provided in the Source Data file. **b** Base quality distribution of the seven ELIGOS datasets at read-level. **c** The normalized current signals from the ELIGOS U/Ψ dataset in GCUCA motif. **d** Visualization of 5-mer signals using Umap under specific sequence contexts (From left to right, they are AGCCA, UGAGU, ACUAA, and UUGUA respectively.). Source data are provided as a Source Data file.

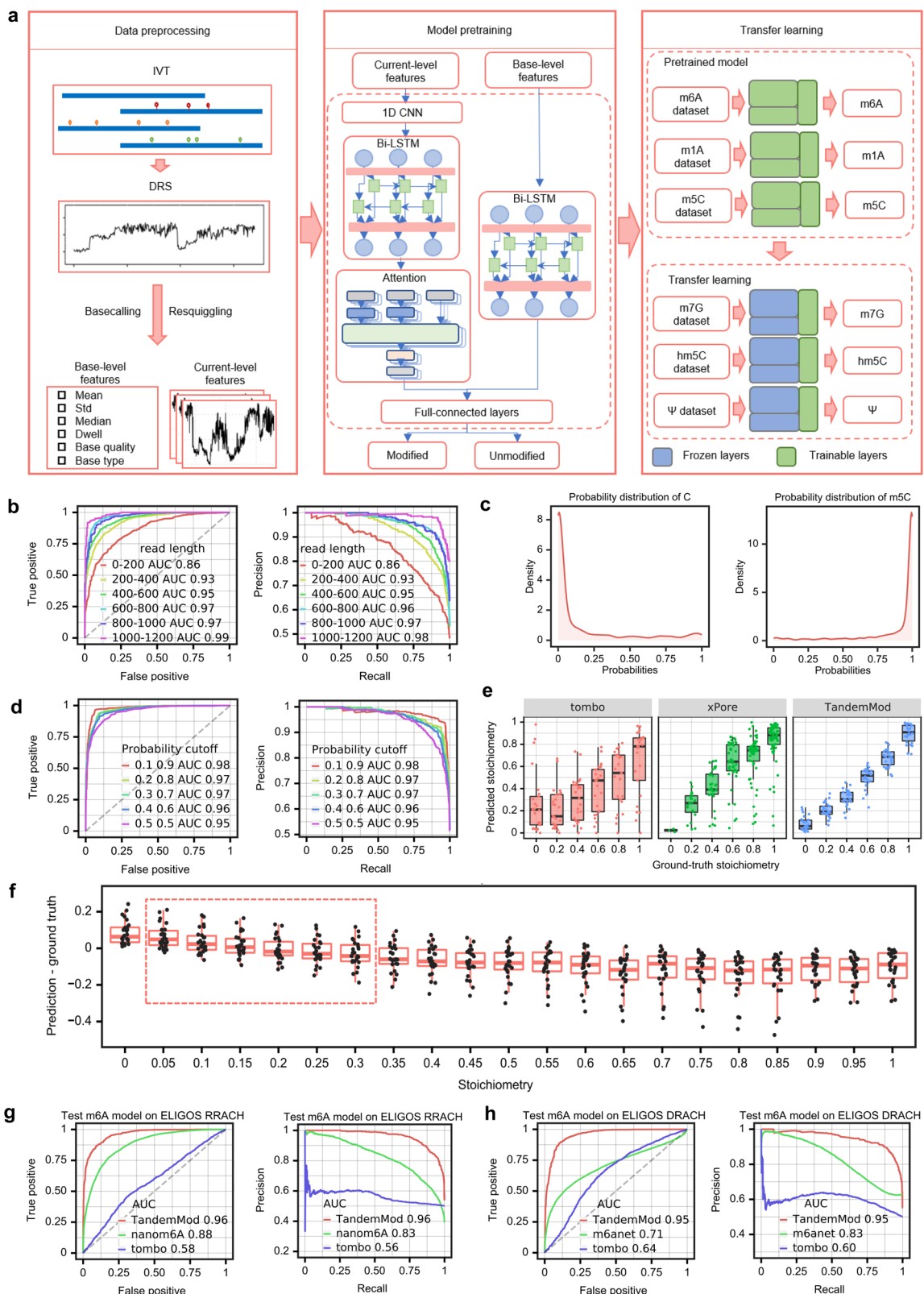

for a more comprehensive and accurate analysis of the presence and distribution of modifications in various biological systems. To achieve this, we developed a deep learning framework, TandemMod, which used 500 features of current signals and 30 base-level features extracted from 5-mer motifs as input (Fig. 2a). TandemMod predicts modifications using features from individual reads, thereby providing information about which reads are modified at a genomic location.

TandemMod consists of 4 main components: a one-dimensional convolutional neural network (1D-CNN), a bi-directional long short-term memory module (bi-LSTM), an attention mechanism, and a classifier comprising full-connected layers (Fig. 2a and Supplementary Fig. 3). The 1D-CNN is utilized to extract local features from raw current intensity signals, while the bi-LSTM module is employed to capture long-term dependencies between adjacent signals. The attention

**Fig. 2 | TandemMod model and performance evaluation on published datasets.**
**a** Schematic of TandemMod model with data preprocessing, model pretraining and transfer learning. **b** ROC curve and PR curve showing the performance for m⁵C identification using reads with different lengths. **c** The distribution of predicted modification probabilities by the TandemMod model on the ELIGOS dataset. **d** Performance for m⁵C identification model on ELIGOS dataset using different cutoff values. **e** Performance comparison of TandemMod, tombo and xPore in data containing 0%, 20%, 40%, 60%, 80%, and 100% m⁵C-modified samples ($n = 2000$). The upper and lower limits represent the 75th and 25th percentiles, respectively, while the center line represents the median; upper and lower whiskers indicate ±1.5× the interquartile range. **f** Boxplot showing the error between the predicted

level and the ground truth level under different stoichiometries ($n = 2000$). The datasets used in this analysis were mixtured from the ELIGOS-m⁵C dataset and the ELIGOS-normal C dataset. The TandemMod m⁵C model achieved low error when predicting sites with modification levels ranging from 0.05 to 0.3. The upper and lower limits represent the 75th and 25th percentiles, respectively, while the center line represents the median; upper and lower whiskers indicate ±1.5× the inter-quartile range. Outliers are not shown in these figures. **g** ROC curve and PR curve showing the read-level performance comparison of TandemMod, nanom6A and tombo in RRACH motifs. **h** ROC curve and PR curve showing the read-level performance comparison of TandemMod, m6anet and tombo in DRACH motifs. Source data are provided as a Source Data file.

mechanism is used to weight the significance of each feature at different time steps, and the classifier is responsible for making predictions based on the combined information from all features. TandemMod provides both prediction labels and their corresponding confident probabilities and can predict modification at both the read level and the site level.

As m⁵C and m⁶A are the most prevalent modifications in eukaryotic mRNAs[1,55,59], we started with m⁵C and m⁶A to explore the performance of TandemMod. We first trained a TandemMod m⁵C model on DRS dataset derived from in vitro transcribed sequences containing all possible 5-mers[36] (denoted as Curlcake dataset). The dataset was divided into training and testing sets at a ratio of 4:1. To improve the generalization performance of the model, we tested three different signal normalization methods by comparing the area under receiver operating characteristic curve (ROC-AUC) on the testing set. From this analysis, we found that the normalization based on median and median absolute deviation (MMAD) displayed the best performance, achieving a ROC-AUC higher than 0.99 on the testing set (Supplementary Fig. 4a). We then evaluated the performance of the trained m⁵C model on an independent ELIGOS m⁵C dataset with a completely different sequence context and the MMAD normalization method obtained a ROC-AUC of 0.95, which was much higher than when using min-max and z-score normalization methods (Supplementary Fig. 4b). We next investigated whether read length would affect the model performance. To do this, unmodified and m⁵C-modified RNAs in vitro transcribed from synthetic Curlcake sequences with different lengths were used to produce DRS data, which were utilized for the prediction by TandemMod m⁵C model. The results showed that the model's performance improved as the read length increased. Keeping reads with a length greater than 600 improved the ROC-AUC from 0.86 to 0.97 (Fig. 2b).

In addition to identifying modification types, TandemMod also provides probability predictions for these modifications. We implemented a probability cutoff strategy in read-level predictions to further improve the performance of the TandemMod model and minimize false positives. First, we examined the distribution of probabilities assigned to Curlcake and independent ELIGOS dataset's read-level predictions. The modified bases are predicted with probabilities close to 1 and the unmodified ones are predicted with probabilities close to 0 (Fig. 2c), indicating that most of the predictions were highly credible. Next, we tested a range of paired probability cutoff values, from 0.5/0.5 to 0.1/0.9, where predictions with probabilities below the first cutoff were considered unmodified, those above the second cutoff were considered modified, and the rest were discarded. From this analysis, we found that when adopting the 0.1/0.9 paired cutoff, approximately 87.5% and 75% of total instances were retained in the Curlcake and ELIGOS datasets, respectively (Supplementary Fig. 4c). We evaluated the performance of our probability cutoff strategy on both Curlcake and ELIGOS m⁵C datasets. Using a probability cutoff of 0.1/0.9 improved the ROC-AUC from 0.99 to 1 in the Curlcake dataset, and from 0.95 to 0.98 in ELIGOS dataset (Fig. 2d and Supplementary Fig. 4d). To investigate whether discarded instances exhibit enrichment for specific sequence motifs, we conducted an analysis on the

proportion of discarded reads using cutoff thresholds of 0.3/0.7 for all 256 5-mer motifs (Supplementary Fig. 5). Additionally, we examined the distribution of predicted modification probabilities for each motif (Supplementary Figs. 6, 7). The results indicate that discarded instances were uniformly distributed across the 256 motifs. Although this probability cutoff strategy resulted in a reduction of total effective reads, the trade-off between precision and recall was acceptable, as the accuracy of retained predictions was significantly improved.

## Superior performance of TandemMod compared to other available tools

We next assessed the performance of the TandemMod model by comparing it to classic machine learning algorithms. To do this, we trained four m⁵C models and four m⁶A models using TandemMod, XGBoost, support vector machine (SVM) and k-nearest neighbor (KNN), respectively, on the same features extracted from the Curlcake training set. The performance of each model on the Curlcake testing set was evaluated based on the accuracy of each individual motif. In the case of m⁶A identification on the Curlcake testing dataset, TandemMod outperformed the other algorithms with an accuracy of 0.90, while XGBoost, SVM and KNN achieved accuracies of 0.84, 0.80, and 0.73, respectively (Supplementary Fig. 4e). Similarly, for m⁵C identification, TandemMod demonstrated best performance with an accuracy of 0.95, compared to XGBoost (0.90), SVM (0.76) and KNN (0.88) on the Curlcake testing dataset (Supplementary Fig. 4f). This comparison highlighted the effectiveness of the deep learning-based model TandemMod in identifying modifications using DRS data.

The modification rate of modified sites from in vivo RNA molecules is dynamic[5] and shows high levels of variation across conditions and samples[39]. To investigate whether TandemMod can accurately identify samples with different levels of modification rate, we generated DRS data with different proportions of modified reads[39]. Specifically, we sampled reads randomly from the ELIGOS dataset to create mixtures with pre-defined stoichiometry of 0%, 20%, 40%, 60%, 80% and 100% of reads with m⁵C sites. We then processed these samples with TandemMod and compared the results to those obtained using tombo[60] and xPore[39]. The performance was evaluated in the form of the consistency of site-wise modification rates predicted by the three tools with the ground truth modification rates. (Fig. 2e). For xPore, the 0-stoichiometry mixture was used as the needed negative control sample. TandemMod successfully identified m⁵C sites in both low-stoichiometry and high-stoichiometry samples. The results showed a strong positive correlation between predicted modification stoichiometry and the ground truth (Pearson $r = 0.956$), outperforming tombo[60] (Pearson $r = 0.495$) and xPore[39] (Pearson $r = 0.712$). As a de novo prediction model, TandemMod required no negative control samples and can accurately predict samples with varying modification rates.

One of the advantages of DRS-based modification detection method is that it can predict nanopore modifications at the read level. To evaluate the read-level performance of TandemMod, we compare the TandemMod m⁶A model against tombo[60], nanom6A[50] and m6Anet[51] using labeled reads from ELIGOS m⁶A dataset[38]. On the ELIGOS RRACH (R−A or G, H−A, C or U) motif, TandemMod, nanom6A

and tombo achieved a ROC-AUC of 0.96, 0.88 and 0.52, respectively (Fig. 2g). On the ELIGOS DRACH (D–A, G, or U) motif, TandemMod achieved a ROC-AUC of 0.95, higher than m6Anet with ROC-AUC of 0.71 and tombo with ROC-AUC of 0.64 (Fig. 2h). These results suggested that, training with in vitro DRS dataset, TandemMod provided the most accurate predictions at the read level among existing tools.

To further control false positive rate, we employ a two-step cutoff strategy to remove low-confidence sites. In the TandemMod model, we added a softmax transformation to the output layer to generate read-level probabilities as well as the prediction labels. In the first step, we employed a read-level probability threshold cutoff to remove low-confidence reads. Then TandemMod aggregated read-level predictions to generate site-level predictions and applied a site-level modification ratio cutoff to further remove false positives. By implementing these two steps, TandemMod ensures a rigorous and effective approach to minimizing false positives in modification detection tasks, resulting in more reliable and accurate predictions. To systematically investigate the model performance on datasets with different proportions of modified reads, we generated mixed samples from ELIGOS data with $m^5C$ ratio ranging from 0 to 1 with a step of 0.05 and evaluated the site-level performance of TandemMod. The site-wise modification rates predicted by TandemMod showed a gradually increase in these samples, consistent with the ground truth (Supplementary Fig. 8a). We then evaluated the error of modification rate between the predicted modification level and ground truth (Fig. 2f). When predicting samples with a modification level of 0, the FPR was ~0.05, and when predicting high-level modification sites, TandemMod exhibited a false negative rate (FNR) of ~0.1. However, when predicting sites with modification levels ranging from 0.05 to 0.3, the predicted modification level closely aligns with the ground truth. This outcome is attributed to the offsetting effect of false positives and false negatives. We further explored TandemMod's performance at site resolution with both balanced and unbalanced datasets, evaluating the impact of site-level cutoff values on the true positive rate (TPR) (Supplementary Fig. 8b–d). The TPRs dropped significantly when the site-level cutoff approached the ground truth modification level across all three mixed samples. This indicates that to achieve a high TPR, the site-level cutoff value should be lower than the true modification rate. We also evaluated the classifier's precision for each site and observed a decrease in precision in samples with a low $m^5C$ rate, alongside a progressive improvement in precision as the modification rate increased (Supplementary Fig. 8e). To explore whether the probability cutoff strategy adapted in this study could improve the model performance on unbalanced data, we adjusted the probability cutoff from the default 0.5-0.5 to 0.1-0.9 and found that the precision significantly improved on these samples (Supplementary Fig. 8f). For the unbalanced sample with a $m^5C$ rate of 0.05, the mean precision improved from 0.37 to 0.65.

## In vitro epitranscriptome dataset with increased sequence diversity improved the prediction accuracy of TandemMod

As mentioned previously, DRS training sets for modification detection can be constructed either using RNAs in vitro transcribed from synthetic sequences or in vivo RNA transcripts[42].

The drawback of RNAs transcribed in vitro from synthetic sequences is the limited diversity of sequence contexts. These sequences are unable to cover all possible sequence contexts, posing a challenge in representing the full range of natural sequences. To address these limitations, we constructed four DRS training sets ($m^1A$, $m^6A$, $m^5C$ and unmodified) by sequencing thousands of transcripts produced from rice cDNA library containing T7 promoter (Fig. 3a and methods), which we termed in vitro epitranscriptome dataset (IVET). Sequencing results for these datasets showed that the sequencing quality scores of the unmodified sample were high, with a mean quality score of over 15 (Supplementary Fig. 9a, b). In contrast, the other 3

modified samples exhibited slightly lower quality scores, which was consistent with the previous findings that modifications decrease the sequencing quality[38] (Fig. 1b). Overall, the sequencing qualities were sufficient for further analysis. In total, transcripts with different contexts from 5260 genes in $m^1A$-modified samples, 4638 genes in $m^6A$-modified samples, 5232 genes in $m^5C$-modified samples, and 3119 genes in unmodified-samples, were detected, respectively. Among them, 2473 genes were in vitro transcribed into RNAs with similar abundance in all four samples (Fig. 3b and Supplementary Fig. 9c). Next, we examined the sequence diversity of k-mers among the IVET, Curlcake and ELIGOS datasets, to determine their coverage of all possible k-mer combinations (Fig. 3c). All three datasets are capable of covering 100% of all possible 5-mers. In the Curlcake and ELIGOS datasets, the coverage of 7-mer motifs decreased to 45.4% and 44.6%, and the coverage of 9-mer motifs decreased to 3.6% and 3.9%, respectively. In contrast, IVET datasets are able to cover all of the 7-mers and 90% of all 9-mers. Furthermore, when we randomly sampled different numbers of genes from IVET and calculated the sequence diversity, we found that when the number of genes reached 1400, nearly 90% of all possible combinations of 9-mer sequences were covered by IVET datasets (Supplementary Fig. 9d).

To construct modification detection models for $m^1A$, $m^6A$ and $m^5C$, we extracted features from the 2473 genes that were common to all four samples in IVET datasets. The datasets were randomly split into training sets (80%) and testing sets (20%). The training sets exhibit a high level of sequence complexity, thereby qualifying them as suitable and representative training resources (Supplementary Fig. 9e). We trained $m^1A$, $m^6A$, and $m^5C$ detection models on their respective training sets and evaluated their performance using ROC-AUC, achieving a range of 0.90 to 0.95 on the IVET testing sets. After applying the probability cutoff previously determined (Fig. 2d), the ROC-AUC improved to 0.97–0.99 (Fig. 3d–f). We further assessed the generalization performance of the trained models on untrained genes by conducting testing on the reserved genes, whose sequences were not encountered during the training phase. The TandemMod models exhibited remarkable precision on the untrained genes (Supplementary Fig. 10a–c), affirming that they can attain high accuracy even in novel sequence contexts. This suggested their potential in identifying modifications in new species. To further investigate whether systematic errors in DRS data caused by different devices would affect model performance, we tested the $m^1A$ model on the ELIGOS dataset and the $m^6A$ and $m^5C$ models on the Curlcake dataset. The IVET datasets were produced using the ONT GridION platform, while the Curlcake datasets were generated using both GridION and MinION platforms. Additionally, the ELIGOS datasets were exclusively generated on the MinION platform. The results showed a slight decrease in accuracy for the $m^6A$ model, while the other two models still maintained high accuracy (Supplementary Fig. 10d–f), suggesting that pretrained TandemMod models can be applied to DRS data generated from different equipment and platforms.

Additionally, we trained TandemMod models on the IVET and Curlcake training sets, respectively, and compared their performance on the ELIGOS dataset. The $m^6A$ model trained on the IVET dataset outperformed the model trained on the Curlcake dataset by 2% (Fig. 3g) in terms of ROC-AUC, while the $m^5C$ model showed a 1% improvement (Fig. 3h). Overall, the performance of TandemMod was further improved when trained on the IVET dataset, demonstrating the potential of the IVET datasets as a valuable resource for training deep learning models to detect RNA modifications.

## Transfer learning of TandemMod for detecting additional types of RNA modifications

Using DRS data, it is possible to achieve high-precision detection of RNA modifications through deep learning. However, this approach requires a large amount of training data that can be costly and time-

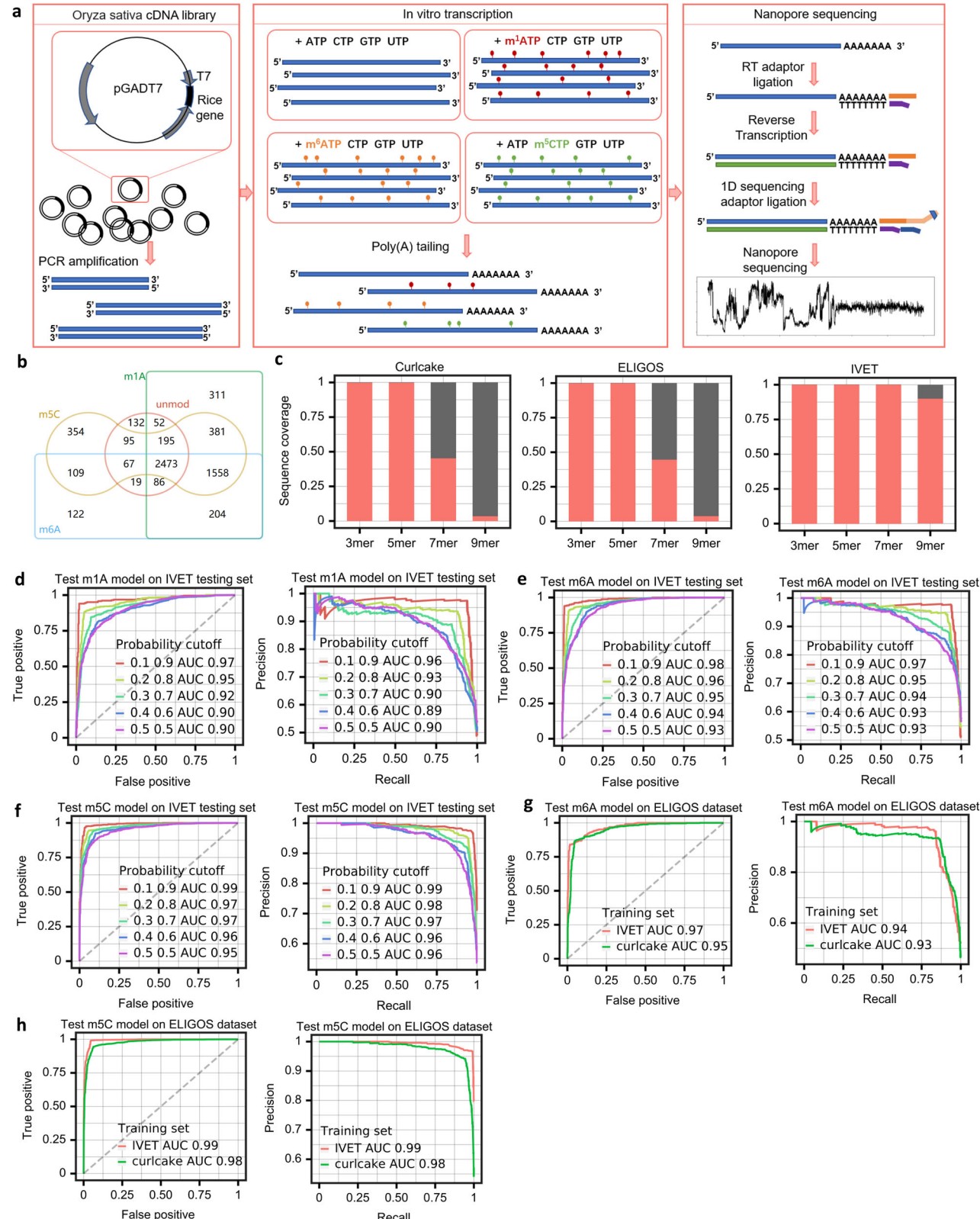

consuming. Therefore, we aimed to develop a cost-efficient and time-efficient method capable of identifying multiple types of RNA modifications using a small amount of DRS data. Transfer learning is a machine learning strategy where a model developed for a task can be reused on a second task with proper adjustment. Transfer leaning has been successfully applied to computer vision and natural language processing and has recently been used to deal with biological

problems[61–65]. Different types of modification detection tasks are similar to each other and are naturally suitable for transfer learning. Therefore, we explored whether transfer learning could be applied to DRS data to achieve the detection for multiple types of RNA modifications. To do this, we trained TandemMod on the IVET m⁵C dataset to obtain a pretrained model. In the TandemMod model, the top layers act as a feature extractor while the bottom layers act as a classifier.

**Fig. 3 | Training TandemMod models on IVET datasets with increased sequence diversity further improved the performance. a** Flowcharts illustrating the construction of IVET datasets. In the in vitro transcription step, canonical nucleotides were replaced with modified nucleotides (m$^1$ATP, m$^6$ATP, and m$^5$CTP). Altogether, three modified samples and one control sample were transcribed from the rice cDNA library. **b** Venn diagram showing the common RNAs among four IVT samples. **c** Bar plots showing sequence coverage in the three datasets in terms of the proportion of motif occurrences in each dataset to all possible 3-mer, 5-mer, 7-mer and 9-mer motifs, respectively. ROC curve and PR curve showing the performance evaluation of the m$^1$A model (**d**), m$^6$A model (**e**) and m$^5$C model (**f**) trained on the IVET training sets and tested on the IVET testing sets, respectively. **g** ROC curve and PR curve showing the performance comparison of m$^6$A models trained on IVET and Curlcake, both tested on the same ELIGOS testing set. **h** ROC curve and PR curve showing the performance comparison of m$^5$C models trained on IVET and Curlcake, both tested on the same ELIGOS testing set. Source data are provided as a Source Data file.

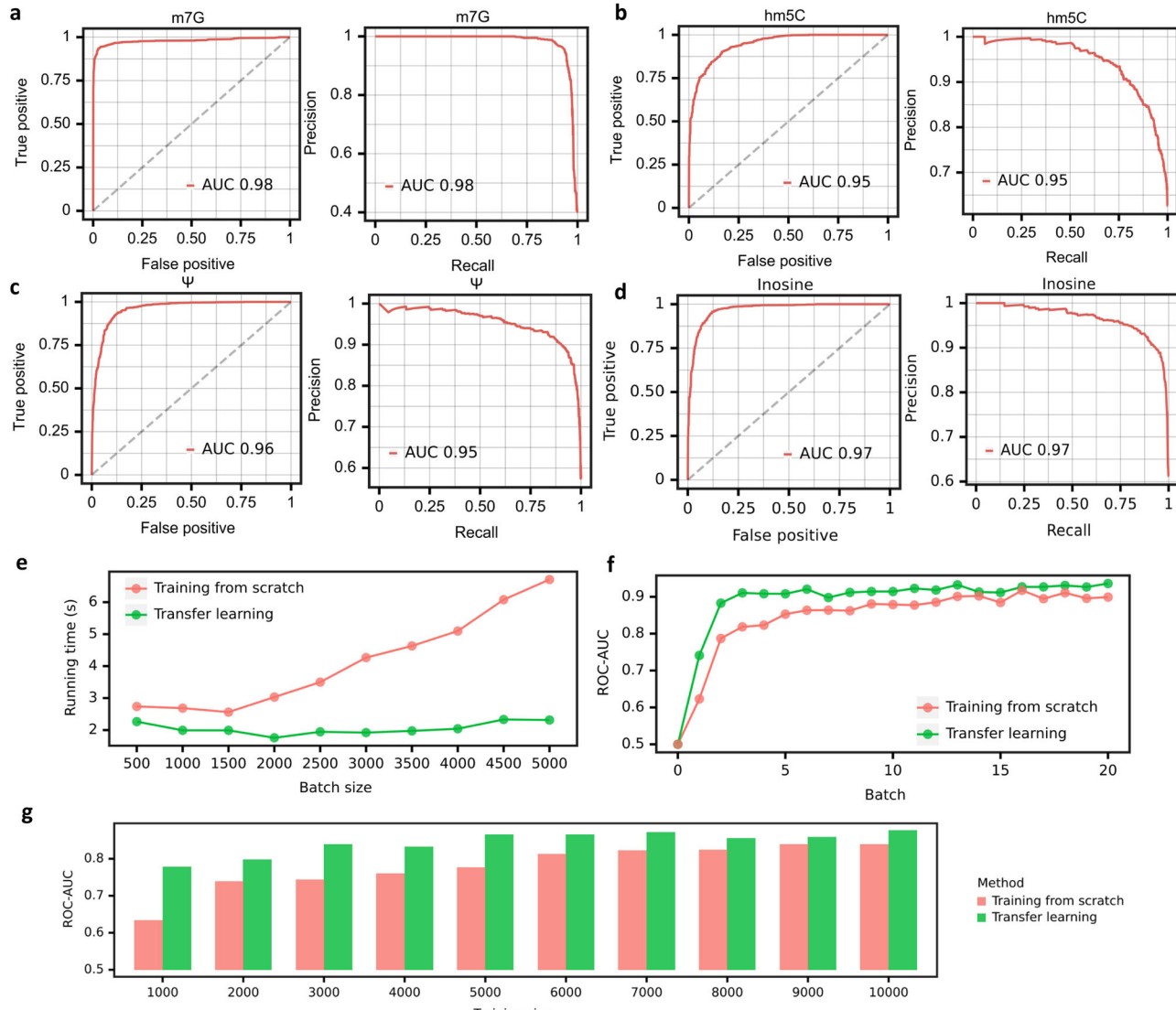

**Fig. 4 | Multiple types of RNA modifications identified by TandemMod.** ROC curve and PR curve showing the performance evaluation of m$^7$G model (**a**), hm$^5$C model (**b**), Ψ model (**c**) and Inosine model (**d**) transferred from m$^5$C model and tested on ELIGOS datasets. **e** Comparison of the running times between the transfer learning mode and training from scratch mode of TandemMod. The running time of transfer learning increases gradually with the increase in batch size, as compared to training from scratch. **f** The ROC-AUC performance of the transfer learning mode and training from scratch mode of TandemMod was evaluated. In the transfer learning process, a pretrained m$^6$A model was retrained on IVET datasets to obtain a m$^5$C model. Meanwhile, the training from scratch process involved training a m$^5$C model directly on the IVET dataset. It was observed that the transfer learning approach achieved higher performance levels more rapidly compared to training from scratch. **g** The ROC-AUC performance comparison between the transfer learning mode and training from scratch mode of TandemMod using different sizes of training data. It is observed that the transfer learning approach achieves higher AUC with less training data. Source data are provided as a Source Data file.

Thus, we froze the top layers of the pretrained model and retrained the bottom layers on ELIGOS training sets (hm$^5$C, m$^7$G, Ψ and I) to minimize the classification error (Fig. 2a). After 2 epochs, all three models achieved high accuracy, with the hm$^5$C model reaching a ROC-AUC of 0.98 (Fig. 4a), the m$^7$G model reaching a ROC-AUC of 0.95 (Fig. 4b), the Ψ model reaching a ROC-AUC of 0.96 (Fig. 4c), and the I model reaching a ROC-AUC of 0.97 (Fig. 4d).

Next, we performed a systematic analysis to quantitively measure the classification performance, required training data, and computational resource utilization of transfer learning from TandemMod m$^5$C

model for m⁶A detection, compared to standard instance of TandemMod m⁶A model. Firstly, we recorded the training time for individual batches under two scenarios: transfer learning with pretrained TandemMod models and training from scratch. The results illustrate a significantly reduced training time per batch in the case of transfer learning compared to de novo training (Fig. 4e). We also evaluated the efficiency regarding the amount of training data required when transferring a pretrained model to a new modification type versus starting the training from scratch. Our findings demonstrate that transferring a pretrained model to new dataset demands considerably less training data without compromising performance (Fig. 4f, g). Furthermore, we assessed the performance of the model across varying training epochs. This evaluation helped us understand the rate at which the model learns new modifications under both transfer learning and de novo training scenarios. The ROC and PR curves indicate that transfer learning not only reduces the computational resources and data requirements but also maintains a high standard of accuracy and efficiency (Fig. 4e, f).

In real-world modification detection tasks, high false positive rate would significantly reduce the reliability and accuracy of the model since the number of unmodified sites is usually much larger than that of modified sites. Thus, to ensure the reliability and accuracy of our models, including m¹A, m⁶A, m⁵C, hm⁵C, m⁷G, I, and Ψ, we evaluated their false positive rates by testing them on the modification-free IVET dataset. The results (Supplementary Fig. 11a) showed that the false positive rates of these models were less than 10% (except for the Ψ model, which was less than 20%), indicating high reliability and accuracy of these models.

To investigate how other types of modifications influence the performance of pre-trained TandemMod on a given modification, we first tested the performance of TandemMod m⁶A model on A sites from the IVET-m⁵C dataset and TandemMod-m⁵C model on C sites from the IVET m⁶A and m¹A datasets. The results showed that the performance of TandemMod m⁶A model was not significantly affected by the upstream or downstream m⁵C sites (from −10 to 10 nt), except for the closest C (+1 nt); Similarly, the performance of TandemMod-m⁵C model on C sites was not attenuated by the upstream or downstream m⁶A or m¹A sites (from −10 to 10 nt) (Supplementary Fig. 12).

Next, we further explored how nearby modifications might influence the predictions using in vivo data. Yeast rRNA molecules are known to contain a variety of well-documented modification sites, and both the proportions and types of these modifications have been extensively annotated[66,67]. To assess how neighboring modifications impact TandemMod predictions, we utilized Direct RNA Sequencing (DRS) data for yeast rRNA. In our analysis, we initially applied the TandemMod m⁵C model to yeast 25 s rRNA, which spans 3389 base pairs, to predict all the C sites. We then calculated the overall FPR for the C sites within the 25s rRNA (Supplementary Fig. 13a, b). Interestingly, we observed that the FPRs for C bases adjacent to modified bases and those adjacent to unmodified bases did not exhibit significant differences. Additionally, we conducted an in-depth analysis of the distribution of predicted modification probabilities for C bases in proximity to modifications. The majority of these predicted modification probabilities were found to be close to 0 (Supplementary Fig. 13c–e), indicating a high level of confidence in their being unmodified. However, we did observe some sites that were somewhat influenced by nearby modifications (Supplementary Fig. 13f), which resulted in a slightly elevated false positive rate. The TandemMod m⁶A model exhibited similar results (Supplementary Fig. 13h–l), further validating the robustness of TandemMod to various other types of modifications. This comprehensive analysis suggests that neighboring modifications in yeast rRNA have a limited impact on the overall false positive rate of TandemMod predictions. This characteristic guarantees that TandemMod is applicable to real-world RNA modifications.

We next investigated the feature extraction ability of the TandemMod model by comparing the input and output features. We visualized the current features of the EILGOS dataset for 6 different modifications before and after being fed into the TandemMod model. The raw current features of the 6 modifications were mixed and difficult to distinguish (Supplementary Fig. 11b). However, after being processed by the feature extractor of TandemMod, the different modifications tended to cluster together in distinct areas (Supplementary Fig. 11c). Specifically, m⁷G and Ψ clustered into separate groups, while m⁵C and hm⁵C were grouped together, and m¹A and m⁶A bases were also grouped together. This indicated that the TandemMod model successfully captured modification-related features while ignoring sequence-related features. This is a significant advantage, as it enables the model to differentiate modifications based on their fundamental patterns, regardless of the sequence contexts. This property allows the model to generalize effectively to new sequence contexts and species without the need for extensive training data. We conducted further evaluations on the feature importance learned by TandemMod regarding the five input bases (Supplementary Fig. 12d and Methods). The results revealed that the TandemMod model places greater emphasis on the centered base compared to the neighboring bases. This observation further supports the notion that TandemMod tends to be less influenced by neighboring modified bases.

In this study, we have demonstrated that TandemMod, through the incorporation of transfer learning, can effectively achieve detection of multiple types of RNA modifications with high accuracy. By leveraging pre-trained models on new tasks, we can improve the performance of our models for detecting specific RNA modifications. The ability of TandemMod to effectively extract relevant features from RNA sequences has led to improved performance compared to traditional machine learning approaches.

## Validation of TandemMod's performance on detecting m⁶A and m⁵C sites in human cell lines

To test whether TandemMod models can be generalized to in vivo DRS data from new species, we used several human cell lines (two modification writer knockout samples and five WT samples) to further validate the reliability of TandemMod. To begin, we used the DRS data from both wild-type (WT) and the RNA methyltransferase METTL3 knockout (KO) HEK293T cells from the Singapore Nanopore-Expression project[68] to identify m⁶A sites. The results showed a significant reduction in the m⁶A/A ratio at both the read and site levels (Fig. 5a, b). As expected, we observed a decreased m⁶A/A ratio at the gene level in METTL3-KO sample compared to the WT sample (Supplementary Fig. 14a). TandemMod predicted A sites within the sequence context NNANN and the sites identified as m⁶A-modified in the WT sample are enriched in canonical DRACH motif (Fig. 5c). Furthermore, the most frequent sequence motif of predicted m⁶A sites was GGACT, consistent with m6Anet results[51]. We then focused on site 1216 of the ACTB transcript and site 1339 of the BSG transcript, which were reported as known m⁶A sites[69]. According to TandemMod predictions, the m⁶A/A ratio at site 1216 of the ACTB transcript was 50.8% and 7.7% in wild-type and METTL3-KO samples, respectively. Furthermore, the m⁶A/A ratio at site 1339 of the BSG transcript was 73.1% and 15.8% in wild-type and METTL3-KO samples, respectively (Fig. 5g). The top 30 most significantly differentially m⁶A-modified genes in METTL3-KO and WT HEK293T cells were further examined (Supplementary Fig. 14c). Among these genes, some were related to RNA processing such as SF3B4, other genes encoded ribosomal proteins such as MRPL54, or served as transcription factors such as ZNF207. To further evaluate the reliability of our method, we applied TandemMod to human lung adenocarcinoma cells (A549), colon cancer cells (HCT116), breast cancer cells (MCF7), liver cancer cells (HEPG2), and leukemia cells (K562)[68] to detect m⁶A sites (Fig. 5h–l). These predicted results consistently matched the DRACH motif, with the most frequent motif

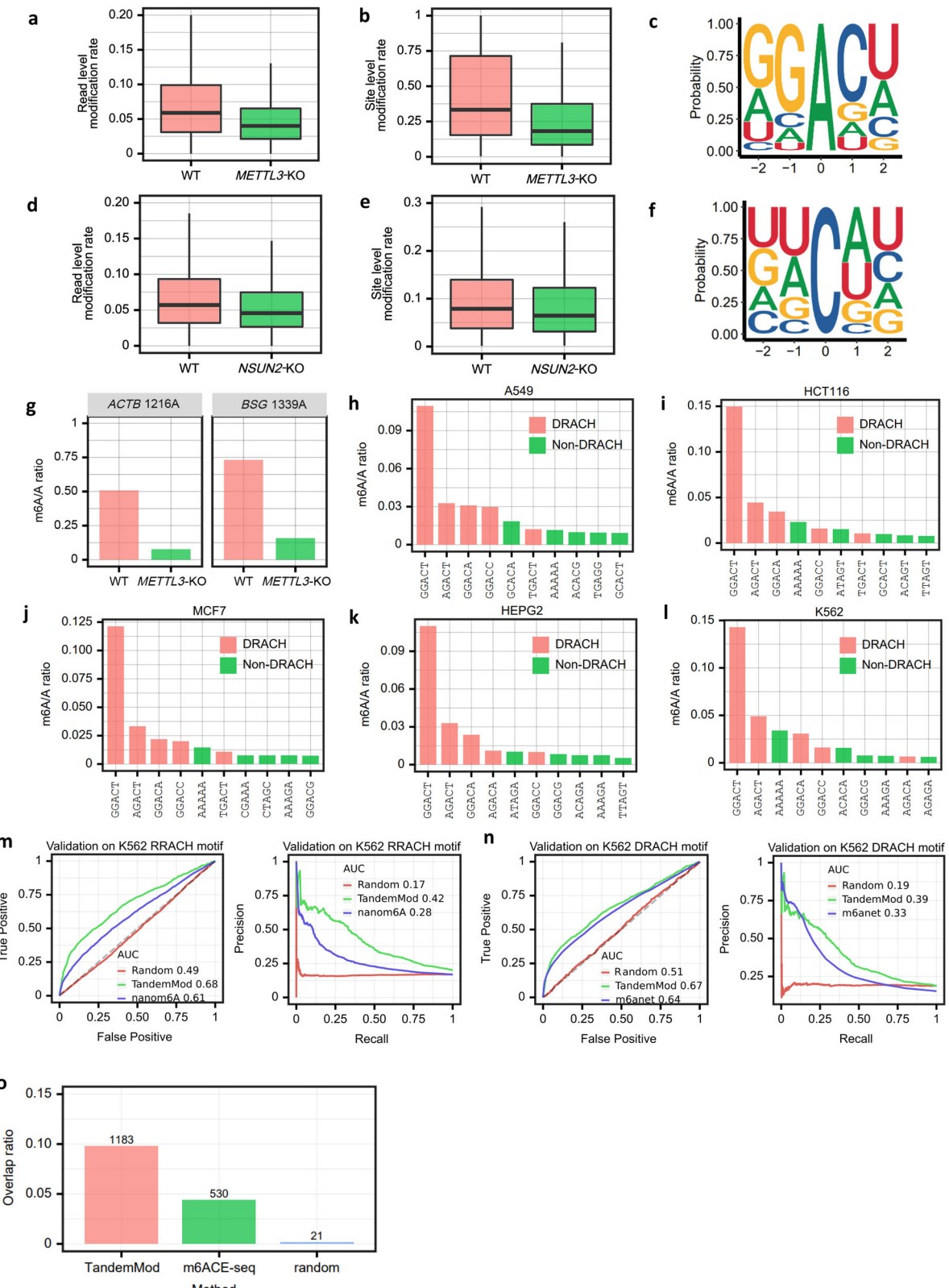

being GGACT, which further validated the reliability of the Tandem-Mod model. We further conducted a comparative analysis of m⁶A levels across different cell lines (Supplementary Fig. 14d), and observed an overall similarity in m⁶A modification rates.

To further evaluate TandemMod performance on in vivo sample, we compared the performance of TandemMod with nanom6A and m6Anet on the detection of m⁶A sites in the aforementioned K562 cell

line. Peaks from the MeRIP-seq[70] of K562 cell line were used as ground truth to evaluate the model performance. As nanom6A was designed for RRACH motifs and m6Anet was designed for DRACH motifs, we extracted RRACH motifs and DRACH motifs from the K562 DRS data, and compared them with nanom6A and m6Anet, respectively. TandemMod achieved a ROC-AUC of 0.69 on RRACH motifs and a ROC-AUC of 0.67 on DRACH motifs (Fig. 5m, n), outperforming both

**Fig. 5 | Validation of TandemMod on various human cell lines. a, b** Modification rate of *METTL3*-KO and WT HEK293T samples at read level ($n = 1520$ for WT, $n = 1860$ for *METTL3* KO) and site level ($n = 11406$ for WT, $n = 15099$ for *METTL3* KO). The upper and lower limits represent the 75th and 25th percentiles, respectively, while the center line represents the median; upper and lower whiskers indicate ±1.5× the interquartile range. Outliers are not shown in these figures. **c** Enriched sequence motif of identified m⁶A sites in the WT HEK293T sample. **d, e** Modification rate of *NSUN2*-KO and WT HeLa samples at read level ($n = 19980$ for WT, $n = 20204$ for *NSUN2* KO) and site level ($n = 76664$ for WT, $n = 86837$ for *NSUN2* KO). The upper and lower limits represent the 75th and 25th percentiles, respectively, while the center line represents the median; upper and lower whiskers indicate ±1.5× the interquartile range. Outliers are not shown in these figures. **f** Enriched sequence motif of identified m⁵C sites in the WT HeLa sample. **g** Modification rates of the known m⁶A-modified sites from *ACTB* and *BSG* mRNA

transcripts predicted by TandemMod in WT and *METTL3*-KO samples. Top 10 m⁶A-modified motifs identified by TandemMod in 5 cell lines A549 (**h**), HCT116 (**i**), MCF7 (**j**), HEPG2 (**k**) and K562 (**l**), respectively. The red bars represent the DRACH motifs. **m** Performance comparison of TandemMod and nanom6A on K562 RRACH motifs. Peaks from MeRIP-seq were used as ground truth. A random classifier which randomly generated a modification probability for each input was used as null control for evaluate the model performance. **n** Performance comparison of TandemMod and m6Anet on K562 DRACH motifs. Peaks from MeRIP-seq were used as ground truth. A random classifier which randomly generated a modification probability for each input was used as null control for evaluate the model performance. **o** Validation of TandemMod on human HEK293T cell line. Sites from miCLIP were used as ground truth. Random sites and sites from m6ACE-seq were utilized for comparison. Source data are provided as a Source Data file.

nanom6A and m6anet. We also compared the predicted m⁶A sites from the HEK293T cell line with m⁶ACE-seq[26] and miCLIP[27] data from the same cell line. We used the 12050 m⁶A sites identified by miCLIP as ground truth, and compared the overlap with TandemMod predictions, m⁶ACE-seq and randomly selected A sites. Out of the reported 12,050 m⁶A sites, TandemMod successfully identified 1,183. Notably, the overlap ratio between TandemMod predictions and miCLIP data was found to be comparatively high when compared to m⁶ACE-seq and randomly selected A sites from DRACH motifs.

We then used DRS data from both WT and the RNA methyltransferase *NSUN2*-KO HeLa cells from bioproject PRJNA872027[71] to identify m⁵C sites. TandemMod also identified significantly decreased m⁵C/C ratio at both read level and site level in the *NSUN2*-KO sample (Fig. 5d, e). We observed a decreased m⁵C/C ratio at the gene level in the *NSUN2*-KO sample (Supplementary Fig. 14b). We observed an UA-rich sequence logo enrichment (Fig. 5f) around differential m⁵C sites, which is consistent with published bisulfite sequencing results[72]. In conclusion, the results demonstrated the effectiveness of TandemMod in accurately identifying modification sites across species and tissues with diverse genetic backgrounds. Its generalization highlighted its potential as a valuable tool for investigating methylation patterns and understanding regulatory mechanisms in various biological contexts.

## TandemMod revealed the epitranscriptome landscape of rice with multiple types of RNA modifications

Current advances revealed that m⁶A, m⁵C, and Ψ are quite abundant in plant transcriptomes[73]. To explore the distribution of multiple RNA modifications on rice transcripts and examine how these modifications respond to environmental stressors, we carried out nanopore sequencing on rice samples subjected to high-salinity conditions (100 mM NaCl treatment), as well as on control (CK) samples under normal conditions and obtained 462,842 and 475,699 DRS reads in NaCl and CK samples, respectively. We employed the TandemMod models to identify m⁶A, m⁵C and Ψ modifications across the entire transcriptome of both treated and control samples. TandemMod identified 26,934 m⁶A sites across 7981 genes, 37,405 m⁵C sites across 6197 genes and 4630 Ψ sites across 2588 genes in CK sample (Fig. 6a, Supplementary Data 1, 2). The majority of modified genes possessed 1–4 modified sites (Supplementary Fig. 15d–g). To validate our findings, we compared the identified m⁶A sites with those from a previous m⁶A-seq study[17]. We found that 70% (5585/7981) of the predicted m⁶A-modified genes and 43.3% (11660/26934) of the predicted m⁶A sites were covered by m⁶A-seq (Supplementary Fig. 15a). Furthermore, to assess whether TandemMod model trained on the IVET datasets could performs better than the model trained on the Curlcake dataset on biological sample. We compared the top 100, 500, 1000, 2000, 5000, and 10,000 predicted m⁶A sites from models trained on IVET and Curlcake datasets with the aforementioned m6A-seq. The analysis showed the top m⁶A sites predicted by TandemMod trained from IVET libraries covered higher proportion of m6A-seq-validated sites as

compared to those predicted by TandemMod trained from Curlcake (Fig. 6b). This finding suggested that the increased sequence complexity in IVET further improved the performance of TandemMod in biological samples. Compared to m⁶A-seq, TandemMod offered predictions with single-base resolution (Fig. 6c), allowing for more precise identification and mapping of exact RNA modification sites in the rice transcriptome.

The predicted m⁶A sites were found to be enriched in the DRACH motif (Fig. 6d left panel). Similarly, the predicted m⁵C sites displayed a UA-rich enrichment (Fig. 6d right panel), which is a known characteristic of m⁵C modification motif. The TandemMod models were trained on all of the motifs, allowing them to discover novel modification motifs. In addition to the DRACH motif, we found that m⁶A modifications were also enriched in the AGA motif based on the prediction results (Supplementary Fig. 15h), which was also revealed by a recent study in plant[74]. This capacity to uncover new motifs can expand our understanding of the diversity and complexity of RNA modification patterns. The distribution of predicted m⁶A and m⁵C sites showed that m⁶A had a preferred distribution near the stop codon and the 3′ UTR region (Fig. 6e left panel) while m⁵C tended to be present near the start codon (Fig. 6e right panel), which was consistent with previous results[75,76] and further validated the reliability of our approach.

Next, we explored which genes tended to contain high frequency of transcripts containing both m⁶A and m⁵C modification in rice under normal condition. To do this, we further identified 4597 mRNAs with high-confidence m⁶A sites (modification rate >0.5) and 3945 mRNAs with high-confidence m⁵C sites (modification rate >0.5). Among these mRNAs, 2394 possess both m⁶A sites and m⁵C sites (Supplementary Fig. 15k). Subsequently, we investigated the frequency of co-occurrence of m⁶A and m⁵C at the same long transcript for each gene. Genes that containing 30% of mRNA transcripts with occurrence of at least one m⁶A and at least one m⁵C were most abundant (Supplementary Fig. 15l). For instance, we identified 58 out of 184 mRNA reads from *LOC_Os03g52840.1* gene contained both one m⁶A and one m⁵C modification, while 7 out of 41 mRNA reads from *LOC_Os03g20700.1* genes showed m⁶A and m⁵C co-occurrence (Fig. 6h, i).

To identify differentially modified sites across the high-salinity and control conditions, we conducted a Chi-square test to compare the m⁶A and m⁵C sites. Genes containing sites with a *p* value less than 0.05 were considered to exhibit differential modification. In total, we identified 363 genes with up-regulated m⁶A-modified sites, and 865 genes with down-regulated m⁶A-modified sites (Supplementary Fig. 15b, c, Supplementary Data 3). Meanwhile, we found 215 genes with up-regulated m⁵C-modified sites, and 1038 genes with down-regulated m⁵C-modified sites (Supplementary Fig. 15i-j, Supplementary Data 4). Gene Ontology (GO) analysis was conducted on all differential m⁶A-modified genes and m⁵C-modified genes in rice revealing significant enrichment in biological processes related to response to abiotic stimuli, various external stimuli, and stress (Fig. 6f, g and Supplementary Fig. 16). The top differentially m⁶A-modified genes and m⁵C-modified genes, highlighting

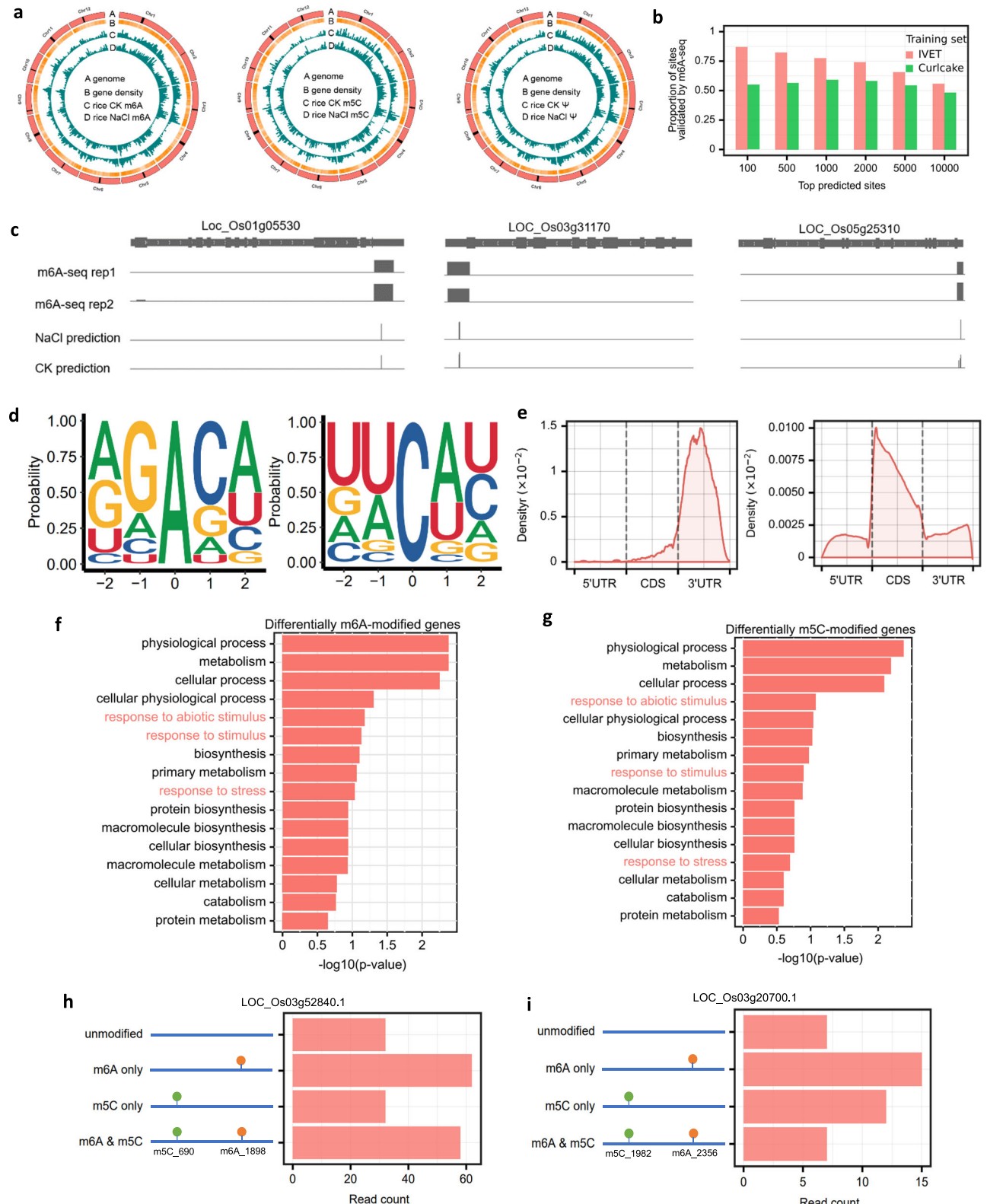

the most significant changes of modification rate in these genes under high-salinity stress. In total, these observations provide further evidence for the importance of m6A and m5C modifications in cellular responses under high-salinity stress.

## Discussion

Nanopore-based modification detection methods face limitations due to the restricted sequence diversity in in vitro synthetic sequences and the absence of reliable labels in in vivo transcribed mRNAs. To address these limitations, we constructed m6A, m1A, and m5C IVET libraries from rice cDNA library producing thousands of transcripts labeled with given modifications for nanopore DRS, providing important resources to serve as benchmark datasets for DRS-based machine-learning methods. Subsequently, we developed a deep learning framework, TandemMod, which is capable of identifying multiple types of RNA modifications in eukaryote transcriptomes, including m6A,

**Fig. 6 | Transcriptome-wide profiling multiple types of rice RNA modifications under normal and high salinity environment. a** Circos plots displaying the density profiles of m⁶A, m⁵C, and Ψ identified by TandemMod in both NaCl-treated and control rice samples. **b** Bar plot showing the proportion of ranked m⁶A sites with high confidence validated by m⁶A-seq. **c** The browser visualization showing three examples of TandemMod-identified m⁶A sites validated by m⁶A-seq in 3′ UTR of given genes. TandemMod was able to detect m⁶A modifications with single-base resolution, which is a significant improvement over traditional m⁶A-seq methods that provide only a limited resolution. **d** Enriched sequence motif of predicted

m⁶A-modified sites (left panel) and m⁵C-modified sites (right panel) in rice control sample. **e** The distribution of predicted m⁶A-modified sites (left panel) and m⁵C-modified sites (right panel) along mRNA transcripts. **f, g** Bar plots show the top 16 enriched Gene Ontology (GO) terms of differentially m⁶A-modified genes and differentially m⁵C-modified genes. Two-sided Fisher's exact test was used without adjusting for multiple comparisons in this analysis. The statistics of reads with simultaneous occurrence of m⁶A and m⁵C from transcripts *LOC_Os03g52840.1* (**h**) and *LOC_Os03g20700.1* (**i**) in the rice WT sample. Source data are provided as a Source Data file.

---

m¹A, m⁵C, hm⁵C, m⁷G, I, and Ψ (Figs. 2–4). For each base, we extracted current intensity with 100 time points and base-level characteristics, including base type, base quality, mean, standard deviation, median and signal length (Fig. 1). TandemMod took the features of the central base and its two flanking bases on each side as input for training. Although mismatch frequencies, insertion frequencies and deletion frequencies can be used to indicate the presence of modifications, this approach cannot identify modifications at the read level. Therefore, these features were not considered in this work. The identification using multiple features associated with modified base and their neighboring bases could potentially enhance the interpretability of the results, and provide valuable insights into the biological implications of the identified modification sites. Our method provides an approach for detecting RNA modifications with high accuracy and without requiring negative control samples, making it a useful tool for RNA modification research.

The accuracy and reliability of TandemMod were thoroughly validated in both synthetic datasets (ELIGOS, Curlcake, and IVET) with ground-truth labels and biological samples (rice and several human cell lines) with labels identified by Illumina-based methods such as m6A-seq, miCLIP and m6ACE-Seq. Furthermore, TandemMod models trained from IVET dataset performed better than those trained from Curlcake both in vitro and in vivo (Fig. 3g, h and Fig. 6b). The superior performance of TandemMod can be attributed to the following factors. First, we constructed IVET datasets for model training, allowing TandemMod models to be trained in various sequence environments and attain wider adaptability. Second, we selected a set of effective features as well as the raw current signals to train TandemMod models, which could effectively capture the differences caused by modifications. Third, TandemMod is a deep learning framework, which is capable of learning complex patterns and relationships in large datasets. The architecture of the TandemMod model allows it to learn intricate patterns and subtle differences in the DRS data that may not be readily apparent to other machine learning algorithms. Taken together, these factors have resulted in TandemMod demonstrating high accuracy and stability, making it an effective modification detection tool. Moreover, TandemMod applied a transfer learning strategy during the training process, which allowed the model to leverage knowledge gained from previous tasks to improve performance on new tasks. This enhanced the adaptability of TandemMod and made it easier to extend to new modifications and species. By fine-tuning the pre-trained model on new datasets, we can achieve better performance with less data and training time, making the models more efficient and effective.

Despite the high performance and adaptability of TandemMod, there are still some limitations that need to be considered. First, its accuracy on short reads may be lower than on longer reads (Fig. 2b). This is because shorter reads tend to have lower sequencing quality at both ends. If the read length is too short, there is a higher likelihood of encountering a proportion of low-quality 5-mers. Therefore, it is recommended to use longer reads when predicting modifications to achieve best performance. To detect RNA modifications on short RNA transcripts like non-coding RNAs, 5′ RNA adapters and 3′ RNA adapters with polyA tails can be attached to the short RNA sequences. This

process extends them into longer ones, which can be used for generating DRS dataset suitable for the precise prediction of modification marks by TandemMod. Second, TandemMod employs a cutoff strategy based on prediction probability and modification rate to improve accuracy. While this strategy can reduce the number of false positives, it may also pose a risk of missing some true positive predictions, especially for modification sites with low sequencing depth. Users should be cautious when interpreting the results and consider alternative strategies for detecting RNA modification in low-depth sequencing samples. TandemMod displayed best performance with DRS signals from long mRNAs (>600 nt) and mRNA with high sequencing depth, while the performance would decrease with current signals from short mRNAs (<600 nt) and mRNA with low sequencing depth. Third, the DRS data, characterized by a significantly larger number of unmodified bases compared to modified ones, are highly unbalanced. When applying TandemMod to detect modification at single base and single-read resolution, the precision of TandemMod is compromised in sites with low modification rate (Supplementary fig. 8e, f). Therefore, although TandemMod can provide modification probability for every single base in single read, only those modification sites detected by TandemMod with high confidence (e.g., modification rate higher than 20% and reads more than 10) were recommended for further examination at read level (single-base and single-read resolution). Finally, when dealing with modifications with high similarity, such as distinguishing hm⁵C from m⁵C, the performance of TandemMod is relatively lower compared to the m⁶A and m⁵C models. With advancements in Nanopore technology, direct RNA sequencing is becoming increasingly accurate, and the performance of TandemMod in distinguishing similar modifications is expected to improve in the future. Although the predictions of TandemMod are generally not influenced by neighboring modified bases, the confidence probability may degrade in some cases (Supplementary Fig. 13f), leading to more discarded reads.

To facilitate the utilization of TandemMod, three different modes has been developed: de novo training, transfer learning, and prediction (https://github.com/yulab2021/TandemMod). The mode of de novo training allows users to train the TandemMod model from scratch using their own datasets. It is ideal for researchers working with novel species or distinct modification types, as it provides the flexibility to build a customized model tailored to their data. In the transfer learning mode, users can fine-tune a pre-trained TandemMod model using their own data. This approach leverages the knowledge gained from the initial training, enabling the model to adapt to new data more quickly and with fewer training samples. In the prediction mode, users can apply a pre-trained or fine-tuned TandemMod model to identify modifications in their dataset. This mode is useful for researchers who want to obtain predictions on modification sites without having to train their own model, saving time and computational resources. By offering these three modes, TandemMod caters to a wide range of research requirements and ensures a user-friendly experience for exploring and analyzing RNA modifications.

In summary, we constructed an in vitro epitranscriptome (IVET) resource using a rice cDNA library, which contains thousands of transcripts labeled with m⁶A, m¹A, and m⁵C modifications. The IVET

dataset, with large sequence diversity and accurate labels, could serve as a valuable resource for machine learning-based RNA modification identification approaches. Then, we developed TandemMod, a transferable deep learning framework capable of accurately identifying multiple types of RNA modifications at single-base resolution in single sample. Overall, the ability to identify multiple types of RNA modifications across species and conditions makes it a useful tool for studying the complex landscape of epitranscriptome in various biological systems.

## Methods

### Plant materials and RNA isolation

Seeds of rice variety DongJing were plated on the Murashige and Skoog (MS) medium (Hopebio, HB8469-5, China) containing 1% sucrose (Solarbio, S8271, China) and 0.3% phytogel (Sigma-Aldrich, P8169, USA), in the presence or absence of salt stress (100 mM NaCl). Two-week-old seedlings (grown at 25 °C, 16-h light and 8-h dark) were harvested and immediately frozen in liquid nitrogen. Total RNA was extracted using RNA isolator (Vazyme, R401-01-AA, China), and treated with RNase-free DNase I (NEB, M0303S, USA) at 37 °C for 15 min to remove any residual DNA. The quality and quantity of RNA was measured by NanoPhotometer NP80 (IMPLEN, Germany).

### In vitro transcription of plant cDNA library and synthetic sequences

The cDNA library containing cDNA of rice seedings and inflorescence in plasmid pGADT7, was commercially available from OE BioTech (Shanghai, China). The T7 promoter and rice cDNAs in the yeast two-hybrid AD library were amplified with Phanta® Max Super-Fidelity DNA Polymerase (Vazyme, P505-d1, China) using PCR primers (AD-preT7-1-F "CTATTCGATGATGAAGATAC" and AD-afterGene-R "GCACGATGCA-CAGTTGAAGT") before and after multiple cloning sites in pGADT7. The PCR program used in the study was as follows: 95 °C for 5 min, followed by 10 cycles of denaturation at 95 °C for 30 s, annealing at 60 °C for 30 s, extension at 72 °C for 4 min, and a final extension at 72 °C for 10 min. PCR products containing T7 promoter and rice cDNA sequences were used as templates for in vitro transcription by MEGAscript T7 Kit (ThermoFisher, AM1333, USA) at 37 °C for 2.5 h to produce non-modified rice mRNAs as control. At the same time, the ATP in the raw material was replaced with $m^1ATP$ (N1-Methyl-ATP – Solid, Jena Bioscience, NU-1027-1) or $m^6ATP$ (N6-Methyl-ATP, Jena Bioscience, NU-1101S), or CTP was replaced with $m^5CTP$ (5-Methyl-CTP, APExBio, B7967) for producing $m^1A$-modiifed RNAs, $m^6A$-modfiied RNA and $m^5C$-modified mRNAs, respectively. We also produced $m^5C$-modified samples from synthetic sequence in this work. The synthetic "Curlcake 1" sequence (2244 bp)[52] used in this study were in vitro transcribed using the AmpliScribe T7-Flash Transcription Kit (Lucigen, ASF3507, USA) with unmodified NTPs to produce control sample; then CTP was replaced with $m^5CTP$ (5-Methyl-CTP, APExBio, B7967) to transcribe $m^5C$-modified sample. The in vitro transcripts were treated with RNase-free DNase I (NEB, M0303S, USA) at 37 °C for 15 min to remove residual DNA, and then were purified by RNeasy Mini Kit (QIAGEN, 74104, USA). The tailing reaction was catalyzed by E. coli Poly(A) Polymerase (NEB, M0276S, USA) at 37 °C for 2 h. The products were purified again by RNeasy Mini Kit, and then ethanol precipitated overnight to obtain RNA for nanopore direct RNA sequencing.

### Nanopore direct RNA sequencing

To prepare sequencing libraries, the ONT's SQK-RNA002 protocol was followed. First, reverse transcription (RT) adaptors were ligated onto RNAs, and then RT was performed at 50 °C for 50 min, followed by 70 °C for 10 min before cooling to 4 °C. The RT products were cleaned up using RNAClean XP beads (Beckman Coulter) and washed with 70% ethanol. RNA motor protein was then ligated onto the RNA strand of the RNA-DNA hybrid, and the reaction was cleaned up with RNAClean

XP beads again before two final washes with WB wash buffer. The library was eluted with EB elution buffer and topped up with RNA running buffer to a final volume of 75 μl before being loaded onto a primed R9.4.1 flow cell (FLO-MIN106D). Sequencing was performed using GridION MK1 and allowed to run for up to 72 h.

### Nanopore signal preprocessing and feature extraction

All IVTs datasets and rice DRS data were processed using the following steps. First, the fast5 files containing raw electric signals were basecalled using Guppy (v6.1.5 with options "--recursive --fast5_out --config rna_r9.4.1_70bps_hac.cfg", available at https://community.nanoporetech.com/). Next, multi-fast5 files were converted to single-fast5 files using ont_fast5_api with default parameters (v4.0.0, available at https://github.com/nanoporetech/ont_fast5_api). Then, base-called sequences were corrected according to reference and aligned to corresponding raw electric signals using tombo resquiggle(v1.5.1 with options '--overwrite --basecall-group Basecall_1D_000 --fit-global-scale --include-event-stdev')[60]. After resquiggling, each 5-mer raw signal was obtained. To eliminate current drift and systematic errors among devices, raw current signals were normalized by median and median absolute deviation. Then, mean, standard deviation, median, signal length and base quality were extracted as features for each base. In order to obtain signals of the same length, each 5-mer raw signal was interpolated using 1-order spline then the 5-mer interpolated signals were resampled to fixed-length (100 in this work). Finally, base-level features and resampled signals of 5 consecutive bases (25 base-level features and 500 resampled current signals) were combined as input of deep learning model.

In nanopore sequencing, RNA/DNA molecules pass through the nanopore at varying speeds, while the sequencing equipment maintains a fixed sample frequency. Consequently, this leads to differing signal lengths for each base. To address this variability, we perform signal resampling to generate signals of uniform length, which are then used as input for our deep learning model. In mathematics, a spline is a special function defined piecewise by polynomials. Assume current signals for a nucleotide with a length of k data points $X = (x_1, x_2, \cdots, x_k)$ were originally sampled from an interval $[a, b]$. Define a function $S$ that maps the singals to $\mathbb{R}$,

$$S : [a, b] \to \mathbb{R} \tag{1}$$

The interval $[a, b]$ can be further divided into k-1 ordered, disjoint subintervals according to each data point

$$[a, b] = [t_0, t_1) \cup [t_1, t_2) \cup \cdots \cup (t_{k-2}, t_{k-1}] \tag{2}$$

For each subinterval, define a polynomial $P_i$ that map the interval to $\mathbb{R}$

$$P_i : [t_i, t_{i+1}] \to \mathbb{R} \tag{3}$$

Then, the original current signals can be represented by piecewise function $P_i$

$$\begin{cases} S(t) = P_0(t), & t_0 \le t < t_1 \\ S(t) = P_1(t), & t_1 \le t < t_2 \\ \quad \vdots \\ S(t) = P_{k-2}(t), & t_{k-2} \le t < t_{k-1} \end{cases} \tag{4}$$

To attain smoothness in these curves, we need to impose constraints that ensure adjacent intervals have the same nth-order

derivatives. These smoothed curves are commonly referred to as splines:

$$\begin{cases} P_{i-1}^{(0)}(t) = P_i^{(0)}(t) \\ P_{i-1}^{(1)}(t) = P_i^{(1)}(t) \\ \qquad \vdots \\ P_{i-1}^{(n)}(t) = P_i^{(n)}(t) \end{cases} \qquad (5)$$

By solving the above constraint equations, we can get the interpolation function $S(t)$. Then, equally divide the interval $[a,b]$ into subintervals $T' = [t'_0, t'_1, \cdots, t'_{l-1}]$ with length $l$ ($l = 100$ in this work) and resample new current values $X'$ from $S(t)$ at these $l$ points.

$$X' = S(T') \qquad (6)$$

In this script, we used 1-order spline interpolation to resample the current signals to fixed length.

## Model design

In this work, the main neural network consists of two sub neural networks, which deal with base-level features and current-level features respectively (Fig. 2a and Supplementary Fig. 3). The first sub-net, which is modified from Song's work[77], consists of a 1-dimensional convolutional neural network (1D-CNN), a bi-directional long short-term memory (bi-LSTM) unit and an attention unit. This module takes resampled signal as input and the context vector of the attention unit as output. The other sub-net is a bi-LSTM unit that takes base-level features (mean, standard deviation, median, signal length, base quality and encoded sequence) as input. The outputs of the two sub-nets are concatenated and input into full-connected layers. The top layers of the model act as feature extractor that learns modification-related information and the bottom layers act as classifier that predict modification type. ReLU activation function is used to perform non-linear operation. Dropout layers are used to mitigate overfitting. Pooling layers are utilized to prune parameters. Cross entropy is chosen as loss function. Adam optimizer is used to update network weights. Cross entropy loss function is defined as,

$$loss = -\sum_{c=1}^{C} w_c \log \frac{\exp(x_{n,c})}{\sum_{i=1}^{c} \exp(x_{n,i})} y_{n,c} \qquad (7)$$

Where $x$ is the input, $y$ is the label, $w$ is the weight, $C$ is the number of classes. In addition to modification labels, the model can also output prediction probabilities. The modification probability is calculated through the softmax of model output.

$$\text{Probability}(y_i) = \frac{\exp(y_i)}{\sum_j \exp(y_j)} \qquad (8)$$

Where $y$ is the output of full-connected layers. Softmax function enables confidence probabilities vary from 0 to 1.

Data with skewed class proportions is very common in real world, e.g., m6A-to-A ratio is typically 0.5% in mammal RNA samples[1]. For data with imbalanced labels, we used a weighed random sampler to reduce the impact of imbalanced data. First, each input data was assigned a weight reciprocal to the sample number of corresponding class. Then, the following random sampler selected data according to weights to ensure balanced data being sampled.

## Model training and evaluation

We trained and tested TandemMod on public datasets (Curlcake and ELIGOS) and in vitro epitranscriptome (IVET) data (m1A, m6A, m5C, and unmodified) with the corresponding modification labels. Each model was trained using an input comprising 800,000 instances from both

modified and control samples, and the test set was composed of 4000 instances. We used the Adam optimizer with a learning rate of 0.001 to train the models. Cross-entropy was used as loss function to update the parameters. The training process was stopped when the model performance on test set was convergent to avoid overfitting. Then, a series of validation tests were conducted to assess the model's performance.

To evaluate the model's fitting ability, we randomly split each dataset into a training set and a test set with a ratio of 4:1, and used the training set for model training and the testing set for model evaluation. The train-test split was performed using a python script located in our repository at https://github.com/yulab2021/TandemMod/tree/master/scripts.

To assess the model's generalization capacity, we set aside a portion of independent reads from select genes as test set, using the reads from remaining genes as the training set. We also performed cross dataset validation to further validate the model's generalization performance. First, TandemMod models were trained on Curlcake datasets and validated on the ELIGOS dataset. Then, TandemMod models were trained on IVET datasets and tested on Curlcake and ELIGOS datasets, respectively. Finally, TandemMod models trained on IVET datasets were validated in two modification writer knock-out human cell lines (METTL3-KO and NSUN2-KO) and five WT cell lines (A549, HCT116, MCF7, HEPG2, and K562).

The classification performance was evaluated by the area under receiver-operating characteristic (ROC) curve and precision-recall (PR) curve. The ROC curve was created by plotting the true positive rate (TPR) and the false positive rate (FPR). The PR curve was created by precision and recall.

$$\text{Precision} = \frac{TP}{TP + FP} \qquad (9)$$

$$\text{Recall} = \frac{TP}{TP + FN} \qquad (10)$$

$$\text{True Postive Rate} = \frac{TP}{TP + FN} \qquad (11)$$

$$\text{False Positive Rate} = \frac{FP}{FP + TN} \qquad (12)$$

Where TP, FP, TN, FN represents true positive, false positive, true negative, false negative, respectively.

## Feature importance evaluation for the five input bases

In the TandemMod model, although features from five consecutive bases are utilized as input, the influence of each base is not uniform. To investigate which base the model predominantly focuses on, we conducted an evaluation of base-level feature importance of the trained TandemMod model. This was achieved by iteratively altering the features corresponding to each of the five bases and comparing the performance against the unaltered, baseline results.

Initially, the TandemMod m6A model trained on the IVET dataset was applied to the ELIGOS m6A dataset to establish the baseline accuracy. Then, for each of the five input bases from the ELIGOS m6A dataset, the base-level and current-level features corresponding to this base were set to 0, and tested with TandemMod to obtain disturbed accuracy. If a particular input base is more important than others, the error between the baseline accuracy and disturbed accuracy will be larger. Thus, the error can serve as a metric of feature importance for the five input bases.

## Control of false positives

In tasks related to modification detection, false positives occur when the model wrongly predicts a site as modified when it's actually unmodified. To tackle this issue, TandemMod employs a two-step strategy designed to control false positives effectively.

Step 1: Adjusting Probability Threshold: In the first step, we employ a read-level probability threshold cutoff. Typically, classification models use a probability threshold of 0.5 to make predictions. However, in TandemMod, we opt for a more conservative threshold of 0.95 to indicate a modified site in our predictions. This higher threshold helps eliminate less confident predictions, significantly reducing the occurrence of false positives.

Step 2: Aggregating Read-Level Predictions: In the second step, TandemMod aggregates read-level predictions to generate site-level predictions. Considering that the DRS datasets with modification events at read level exhibit imbalance, and sequence depth significantly affects the precision and recall of perdition tools[78], specific criteria must be met for a site to be classified as a true modified site: it should have more than 10 supporting reads, and the modification rate must be at least 0.2. This step further contributes to the reduction of false positives.

By implementing these two steps, TandemMod ensures a rigorous and effective approach to minimizing false positives in modification detection tasks, resulting in more reliable and accurate predictions.

## Read-level predictions and site-level predictions

TandemMod is designed to provide highly detailed information about modifications in RNA. It achieves this by taking individual reads as input and predicting modifications on a per-read basis. Furthermore, TandemMod goes a step further by providing confident probabilities for these predictions. To calculate these probabilities, TandemMod employs a softmax transformation. This transformation is applied to the output layer's vector, where each element corresponds to a specific class or modification type. The softmax function ensures that the probabilities sum up to 1 for all classes.

$$p_i = \frac{e^{z_i}}{\sum_{j=1}^{N} e^{z_j}} \tag{13}$$

Where $p_i$ is the confident probability for each class $i$. $z_i$ is the neuron output for each class $i$. $N$ is the number of total classes.

TandemMod aggregates all available reads within a specific genomic location to calculate the modification rate for that location. For more in-depth information and details, please refer to our online documentation at https://yulab2021.github.io/TandemMod_document for a complete guide on how TandemMod works.

## Transfer learning

TandemMod used transfer learning to enhance its performance in detecting specific RNA modifications. Specifically, it fine-tuned pre-trained neural network models on DRS data to detect specific modifications. The pre-trained models were initially trained on IVET datasets to learn general features. During transfer learning, we first froze the top layers of the pre-trained model, including convolutional and pooling layers, to preserve its ability to extract sequence features. Then, the parameters of the fully connected layers of the pre-trained model were updated through backpropagation using Adam optimizer and cross-entropy loss function. Early stopping was employed to prevent overfitting, and the best performing model is selected based on the ROC-AUC score on the validation set. TandemMod can also employ transfer learning across different species. By fine-tuning a pre-trained model on DRS data from one species, TandemMod can detect modifications in another species without requiring a large amount of training data. This allows TandemMod to be more widely applicable for modification detection in different organisms.

## Comparison between TandeMod and other tools on human cell lines

To measure the relative performance of TandemMod and other models. We tested the performance of TandemMod on human K562 cell line and compared with nanom6A and m6Anet. The DRS data for K562 cell line was obtained from Singapore nanopore expression project[68]. To have a fair comparison with these tools, DRACH motifs and RRACH motifs were utilized for comparison with m6Anet and nanom6A, respectively. MeRIP-seq[70] from the same cell line was used as ground truth and ROC-AUC and PR-AUC were used as metrics to evaluate the model performance. It is important to know that, owing to both computational methods and experimental method have their own bias and the in vivo modification state is dynamic, the absolute overlap between different methods may not be very high, however, the relative performance is still comparable. The comparison of TandemMod with miCLIP[27] and m6ACE-seq[26] was conducted on HEK293T cell line. The DRS data was obtained from Singapore nanopore expression project[68]. The overlap between TandemMod predictions and miCLIP was compared to that of m6ACE-seq and randomly selected A sites from DRACH motifs with miCLIP.

## De novo prediction of multi-types of modifications in rice mRNAs

To evaluate the response of mRNA modifications to salt stress, mRNAs from rice seedling under normal condition and salt stress (100 mM NaCl) were used for nanopore sequencing. To identify multi-types of RNA modifications transcriptome-wide at single base resolution, we first extracted base-level and current-level features from both NaCl treated and control rice samples. Different types of RNA modifications, including m6A, m5C, and Ψ, were identified by TandemMod models. To control false positives, sites in each read with confident probability >0.95 were considered as modified and sites in each genomic locations with more than 10 reads and minimal modification rate of 0.2 were considered as true modified sites. For conducting differential modification analysis, we employ Fisher's exact test, a variant of the Chi-square test method. In this analysis, we compare two samples to identify genes that exhibit differential modifications. Each gene's modification count and expression count in the two samples are used to construct a 2 × 2 contingency table. A distinct Fisher's exact test is performed for each gene, with each test focused on a specific pairwise comparison.

| 2 × 2 contingency table for one gene | | |
|---|---|---|
|  | Total gene count | Modified gene count |
| CK | a | b |
| NaCl | c | d |

The $p$ value of Fisher's exact test is:

$$p = \frac{\text{binom}(a+b,a) \times \text{binom}(c+d,c)}{\text{binom}(a+b+c+d,a+c)} \tag{14}$$

Sites with Chi-square $p$ value less than 0.05 were considered as significantly salt-sensitive. Genes that contain salt-sensitive sites were selected for GO enrichment analysis using RiceNetDB (https://bis.zju.edu.cn/ricenetdb).

## Reporting summary

Further information on research design is available in the Nature Portfolio Reporting Summary linked to this article.

## Data availability

Published DRS datasets used in this study were obtained from the work of Novoa lab in National Center for Biotechnology Information (NCBI) under the accession number GSE124309[52], PRJNA563591[36] and the work of Intawat Nookaew under accession number SRP166020[38]. DRS data of several human cell lines used in this study were publicly available from Singapore nanopore expression project, which are available at European Nucleotide Archive (ENA) under the accession number PRJEB44348[68]. Rice m⁶A-seq data was obtained from Gene Expression Omnibus (GEO) under the accession number GSE135549[17]. The m6ACE-Seq and miCLIP data for human HEK293T cell line were available in GEO database under the accession numbers of GSE124509[26] and GSE63753[27], respectively. The MeRIP-seq for human K562 cell line was available in GEO database under the accession number of GSE205709[70]. Four In Vitro EpiTranscriptome (IVET) datasets, two IVT datasets derived from synthetic sequences and two rice datasets generated by ONT Direct RNA sequencing have been deposited in GEO database under the accession number GSE227087. Source data are provided with this paper.

## Code availability

The source code of the TandemMod[79] is available for research purposes at Github: https://github.com/yulab2021/TandemMod (https://doi.org/10.5281/zenodo.10901797). Online documentation and run examples are available at https://yulab2021.github.io/TandemMod_document. The code to reproduce results in this manuscript is available at https://github.com/yulab2021/TandemMod/tree/master/results_reproduce.

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

## Acknowledgements

This work was funded by grants from Shanghai Pujiang Program (Grant No. 21PJ1407600 to X.Y. and 22PJ1406200 to X.L.), National Natural Science Foundation of China (Grant No. 32170581 to X.Y.), and the Science and Technology Commission of Shanghai Municipality (Grant No. 22JC1401300 to H.W.).

## Author contributions

X.Y. and Y.W. design this study. Y.W. and P.X. performed the computational analysis. W.S. performed the IVET experiments. G.H. provided the cDNA libraries. H.W. J.Y., M.Y. and Y.Q.W. provided the library construction and direct RNA sequencing using ONT. X.Y., B.D.G, X.L. and

Y.W. wrote the manuscript. All authors reviewed and approved the final version of the manuscript.

## Competing interests

The authors declare no competing interests.
