## [Peer Review File · Nature Communications]

Transfer learning enables identification of multiple types of RNA modifications using nanopore direct RNA sequencingReviewer #1 (Remarks to the Author):

In this study, Wu and colleagues introduce TandemMod, a machine learning based approach to profile different RNA modifications from direct RNA-seq data. The Authors claim that this framework improves previously proposed approaches both in terms of training data acquisition, and software implementation. Specifically:

- Wu et al use actual transcripts as templates to produce RNA molecules, either fully modified or unmodified, which constitute TandemMod's training sets. These molecules, in contrast to synthetic RNAs previously used for this purpose such as Curlicakes, more accurately depict the complexity of real RNA sequences.
- The Authors propose a deep-learning method which they claim outperforms alternative tools for RNA modifications profiling in terms of predictions accuracy. Moreover, they suggest that TandemMod's design, which is based on the transferable deep learning paradigm, enhances the algorithm re-training for different RNA modifications, and its applicability across conditions.

The paper is overall well written, TandemMod's documentation looks detailed and comprehensive, and the topic is timely and interesting. Indeed, recent work has highlighted substantial room for improvement in detecting RNA modifications from dRNA-seq data (refer to <https://doi.org/10.1038/s41467-023-37596-5>), while the simultaneous profiling of multiple RNA modifications is a promising yet understudied field of research.

Despite these positive aspects, I have several major concerns about the work by Wu et al., which I believe should be addressed before publication.

Major Comments:

1) The Authors highlight an intriguing application of TandemMod in profiling multiple RNA modifications within the same dRNA-seq dataset for a more comprehensive characterization of the epitranscriptome. This concept emerges from the title of the manuscript, and it is immediately mentioned in the abstract. However, in the manuscript, the Authors only marginally comment on the presence of different marks on the same transcript despite TandemMod should be in principle suitable for this kind of analysis. The Authors should elaborate on this aspect in the results section, exploring the simultaneous occurrence of different marks at the single transcript resolution. Moreover, the Authors claim that detecting multiple RNA modifications simultaneously with Illumina-based methods is unfeasible (Line 47), but simultaneous profiling of m5C, Ψ , and m1A has been achieved using RBS-Seq (refer to <https://doi.org/10.1073/pnas.1817334116>). I recommend acknowledging this study and comparing TandemMod's results in HeLa cells with those obtained using RBS-Seq on the same cell line, as reported in the aforementioned paper. Additionally, the Authors perform an analysis on rRNAs, stating it sufficiently demonstrates that detecting one mark with TandemMod is not influenced by the presence of other RNA modifications. To strengthen this conclusion, I suggest running TandemMod, following its training on m6A, to detect m5C and PseudoU marks on IVET datasets to evaluate the impact of these modifications. The analysis could be repeated for all three RNA modifications. Moreover, while False Positive Rate (FPR) is crucial, assessing the impact of nearby RNA modifications on TandemMod's True Positive Rate (TPR) is equally essential to affirm that RNA modification proximity does not affect TandemMod.

2) The Authors present TandemMod as a tool suitable to profile RNA modifications at single base and single read resolution, i.e. a modification probability for sequenced base, and they support their claim based on the analysis of synthetic RNAs. The tool achieves median FP and TP rates of ~5% and ~90% respectively for m5C (Figure2G) and comparable FPR for other RNA modifications (Figure4D). These results are promising, nevertheless, I am not convinced they are good enough to claim such a fine resolution.

I urge the Authors to extensively discuss the Precision of a classifier with a 5% FPR when applied to a dataset characterized by a significantly larger number of unmodified sites compared to modified ones (Line 301). They should demonstrate that TandemMod's FPRs for claimed RNA modifications ensure sufficient precision based on the expected stoichiometries of the marks.

A relevant yet indirect observation in this regard, it is surprising to me that m6A (the most abundant internal methylation in mRNAs) and m5C have the same "Read level modification rate" distribution (Figure 5A-D), which is also perfectly in line with the one presented in Figure 2G for the 0-stoichiometry condition (only driven by false positives).

Despite the single base and single read resolution, TandemMod also provides a functionality to merge the information of reads spanning the same genomic location. As the Authors mention, this is extremely important to control false positives as clearly emerges comparing Figure 5B-E against Figure 5A-D. Due to the relevance of this point, I would move the description of this procedure from the methods to the results session, and I would include the analysis of TandemMod's performance at site resolution, with both balanced and unbalanced datasets, and an exhaustive discussion of parameters tuning as the Authors did for the read-level probability threshold.

3) The Authors comment about the ambiguity of antibody-based methods (Line 66) but their claim is not clear to me. This is extremely important since, based on this sentence, they decided to avoid a comparison with Illumina based methodologies to check the reliability of TandemMod predictions (except for a limited comparison with the rather old m6A-seq technique) relying almost exclusively on synthetic RNAs. Synthetic datasets are very informative but they lack important confounding factors: real sequence complexity, RNA modifications stoichiometry, gene expression levels heterogeneity, and other RNA marks. While several of these aspects have been partially addressed by the Authors (see also previous points), I strongly suggest extending the characterization of TandemMod's performance to real biological datasets also profiled with state of the art Illumina based methods. A great asset in this regard is the work recently published on Nature Communications by Zhong et al (<https://doi.org/10.1038/s41467-023-37596-5>) where 10 tools for m6A profiling were benchmarked against each other's and recent orthogonal Illumina techniques. Leveraging on the data and analyses presented by Zhong et al, the Authors could deeply characterize the performance of TandemMod in presence of all the relevant confounding factors previously mentioned. In my opinion, this analysis is mandatory also to support the claim that TandemMod is superior to existing tools for RNA modifications detection which is so far limited to few other methods compared in the context of synthetic RNAs (Line 226). Finally, the reported improvement in performance due to the sequence complexity retained by IVET compared to Eligos or Curlcake IVTs is minor, however, this could emerge more strikingly when estimating performances based on real biological datasets and Illumina orthogonal techniques.

4) The Authors claim that TandemMod's design allows to leverage on a version of the algorithm trained on a specific RNA modification to learn how to classify a different mark more easily in terms of amount of required data and computational resource. However, I believe this session of the results is too qualitative. I recommend conducting a systematic analysis to measure classification performance, required training data, and computational resources for a standard instance of TandemMod designed for m6A detection compared to a counterpart trained using top layers trained on m5C. The Authors should quantitatively discuss how much TandemMod's design facilitates this purpose.

Minor Comments:

5) Line 94: It would be interesting to extend analyses to all available RNA modifications from the dataset released by Jenjaroenpun et al., especially considering the significance of Inosine in the field.

6) Line 119: Could the Authors expand on their considerations regarding the relative differences in current intensity between modified bases and their unmodified counterparts or other canonical nucleotides?

7) Figure 2B: The PR curves should include the expected performance of the null model, represented by a horizontal line with precision equal to the positive rate. Additionally, it would be useful to graphically depict the expected precision and recall for the default analysis setup.

8) Figure 2E/Line 179: Does the filter on probabilities uniformly affect all 5-mers? It could be informative to present a plot similar to Figure 2E for each sequence context. Moreover, are the discarded instances enriched for specific sites?

9) Line 211: Did the Authors test xPore on the 0-stoichiometry condition using two independent sets of unmodified reads?

- 10) Lines 254-258: How does subsampling affect the number of genes represented in the training dataset and, consequently, the represented sequence complexity?
- 11) Line 272: The concept of "cross-device" applicability requires clarification. Does this conclusion depend on the sequencing platform used to collect the different datasets (e.g., MinION vs. GridION)?
- 12) Extended Figure 6B: The color scale is missing.
- 13) Typos: Line 26: sties -> sites; Line 929: i -> bold "i"

Reviewer #2 (Remarks to the Author):

I co-reviewed this manuscript with one of the reviewers who provided the listed reports as part of the Nature Communications initiative to facilitate training in peer review and appropriate recognition for co-reviewers.

Reviewer #3 (Remarks to the Author):

This manuscript by Wu et al authors proposed a robust deep learning framework for identification of multiple types of RNA modifications from direct RNA sequencing data. Their model was trained based on carefully designed dataset with sufficient sequence diversity, which greatly addressed the major limitation of IVT-based methods. They also performed systematic tests with both in vitro and in vivo transcribed dataset. The work is very timely. I believe there are multiple research groups working on essentially the same projects.

I have the following comments.

1. Although authors used cDNA library as template to get RNA molecules of diverse sequence diversity, the generated dataset may not perfectly simulate the real modification status. For example, in 5-mer 'AAAAA', all the five adenosine will be completely replaced by modified bases into "66666", how this issue is addressed?
2. Authors performed comprehensive comparisons with basic machine learning methods, but not state-of-art methods. ONT-based dictation of RNA modifications is a very hot topic. There are new tools published monthly. I guess it is probably impossible to compare with the best existing approaches published in the past year, please try to get them cited in the introduction section. For example, DENA [1] and Penguin [2]. I am sure there are many more high impact works in this topic that should be cited at least, if not compared.
3. Related to point 1, validation on human cell lines seems a little weak, not sufficient or strong. Specifically,
 - a. The performance for m5C modification does not look good. According to figure 5d-5e, the predicted results for WT and NSUN2-KO are not significantly differed, the difference in site-level results is even smaller than read-level, which looks weird.
 - b. Also, if authors want to demonstrate the utility of their model on biological samples, they should compare different tools. For example, Tombo and ELIGOS for non-canonical base detection, m6Anet and nanom6A for m6A detection, etc.
4. Regarding signal resampling (Line 586), the 100 signal points are randomly sampled or follow a time sequence order? Please clarify.
5. My biggest concern is about the main contribution of the manuscript. Performance wise, it is always difficult to claim it is the best, with new methods being published all the time. The main contribution of the work is its capability of predicting the most types of RNA modifications. Only six different types of RNA modifications were supported, which seems rather limited given that there are at least twelve different types of widely occurring RNA modification according to a previous work (m6A, m1A, m5C, m5U, m6Am, m7G, Ψ, I, Am, Cm, Gm, and Um). Please consider

supporting the prediction of A-to-I (or other modification type) in the model.

Reference

1. Qin, H., et al., DENA: training an authentic neural network model using Nanopore sequencing data of Arabidopsis transcripts for detection and quantification of N(6)-methyladenosine on RNA. *Genome Biol*, 2022. 23(1): p. 25.
2. Hassan, D., et al., Penguin: A tool for predicting pseudouridine sites in direct RNA nanopore sequencing data. *Methods*, 2022. 203: p. 478-487.

Reviewer #4 (Remarks to the Author):

TandemMod is a novel deep learning framework that allows identification of multiple RNA modifications in Nanopore sequencing data. TandemMod's ability to characterize RNA modifications at the read level and lack of requirement for control samples are significant improvements in Nanopore-based detection of modifications. This study represents an important step in moving to the next "phase" of epitranscriptomics research where scientists can investigate the effect of combination of RNA modifications on biology. To make TandemMod useful for the epitranscriptomics community, the authors developed several modes for TandemMod utilization. The authors present evidence necessary to support their conclusions, however, several points in the manuscripts needs to be addressed prior to publication. I am in favor of publication of this study if the raised issues are addressed by the authors.

L.39: adding "transgenic" before "human" would make it more clear for the reader

L.65 and 66: it is not clear which antibody-based immunoprecipitated experiments are being referred to here. Reference as well as a little further explanation is necessary.

L.72: It is not clear why these specific modifications were chosen for IVET dataset. The researchers are investigating more than these three modification in the rest of the manuscript, so it is unclear here as well as later in the text why they are concentrating on these A-based modifications.

L.75: Can TandemMod be used to identify multiple modifications in the same read?

L.92: Are the "in vitro-transcribed datasets" ELIGOS datasets? If yes, this needs to be clearly indicated for the reader. As ELIGOS are described in detail later in the text, this appears as a different dataset.

L. 106: In this paragraph the authors are describing how different parameters can be used to differentiate the 6 RNA modifications in the study. The data is presented in the figure, however, for the reader, it might be beneficial to also present this in a table where it is indicated exactly which parameter combination allows differentiation of the 6 modifications.

L.108: Could the authors explain what they mean by "resquigging" as this might not be the term that the audience of Nature Communications is widely familiar with?

L.110-111: Could the authors explain what they mean by signal re-sampling with spline interpolation?

L. 116: Could the authors explain what 2D Umap space shows/indicates? Additionally, how does the specific sequence around the modified nucleotide contribute to clustering?

L.202-226: For the experiments described in these paragraphs, it is not clear why these specific modifications (m5C and m6A) were selected for each of these? It is not clear whether the same result would be expected for the other modifications mentioned in the manuscript.

L. 236: Construction of IVET database (Figure 3a) needs to be described in more details in the methods section. For example, in L. 533 it is not clear how the cDNA pGADT7 library was constructed. Is it a commercially available library? If so, please refer to where it can be obtained. If the researchers generated it themselves, methods need to be updated with this information. For the experiments generating IVET training sets, how would incomplete replacement of ATP (i.e., when let's say 50% of ATP is unmodified) affect training of the model and identification of modification? Does complete replacement of unmodified nucleotide with modified nucleotide limit the diversity of sequences that are used for training of the data?

L. 272: It is not clear what the authors mean by "cross-device" applicability.

L.336: Not all of the modifications clustered in distinct areas in Fig. 4f. Could the author comment on what consequences this lack of distinct clustering for some modifications has in terms of using TandemMod in the future?

L.365: Why were these m6A sites specifically reliable?

L. 368: Why were top 30 gene selected? How do these identified genes compared to the previously published data on m6A-epitranscriptome? I think in order to demonstrate that the method is working in vivo, the authors need to perform a more thorough comparison to the available data. Identification of the DRACH motif should not be the only measure of reliability assessment. This point also applied to the m5C data in the next paragraph (L.380-390).

L. 483: In the discussion pertaining to accuracy on prediction on short reads, could the authors comment on the subsets of biologically relevant RNAs that TandemMod might not be applicable to (for example, small non-coding RNAs)? Is there a way to modify TandemMod in the future to make it work on short reads?

Point-by-point response (re: Manuscript NCOMMS-23-54139)

Editor's comments

Thank you again for submitting your manuscript "Transferable deep learning enables identification of multiple types of RNA modifications using nanopore direct RNA sequencing" to Nature Communications. We have now received reports from 4 reviewers (2 of which were co-reviewers) and, after careful consideration, we have decided to invite a major revision of the manuscript.

As you will see from the reports copied below, while the reviewers find your deep learning model TandemMod to be of potential interest, they also raise important concerns. Please address all the reviewer's comments in your revision, while bearing in mind that we will be reluctant to approach the referees again in the absence of revisions.

REVIEWER COMMENTS

Reviewer #1 (Remarks to the Author):

In this study, Wu and colleagues introduce TandemMod, a machine learning based approach to profile different RNA modifications from direct RNA-seq data. The Authors claim that this framework improves previously proposed approaches both in terms of training data acquisition, and software implementation. Specifically:

- Wu et al use actual transcripts as templates to produce RNA molecules, either fully modified or unmodified, which constitute TandemMod's training sets. These molecules, in contrast to synthetic RNAs previously used for this purpose such as Curlcakes, more accurately depict the complexity of real RNA sequences.

- The Authors propose a deep-learning method which they claim outperforms alternative tools for RNA modifications profiling in terms of predictions accuracy. Moreover, they suggest that TandemMod's design, which is based on the transferable deep learning paradigm, enhances the algorithm re-training for different RNA modifications, and its applicability across conditions. The paper is overall well written, TandemMod's documentation looks detailed and comprehensive, and the topic is timely and interesting. Indeed, recent work has highlighted substantial room for improvement in detecting RNA modifications from dRNA-seq data (refer to <https://doi.org/10.1038/s41467-023->

37596-5), while the simultaneous profiling of multiple RNA modifications is a promising yet understudied field of research.

We thank the Reviewer very much for all these positive comments and thorough review of our work. In the revised manuscript, we have addressed all the comments and summarized new data and discussions suggested by the Reviewer.

Despite these positive aspects, I have several major concerns about the work by Wu et al., which I believe should be addressed before publication.

Major Comments:

1) The Authors highlight an intriguing application of TandemMod in profiling multiple RNA modifications within the same dRNA-seq dataset for a more comprehensive characterization of the epitranscriptome. This concept emerges from the title of the manuscript, and it is immediately mentioned in the abstract. However, in the manuscript, the Authors only marginally comment on the presence of different marks on the same transcript despite TandemMod should be in principle suitable for this kind of analysis. The Authors should elaborate on this aspect in the results section, exploring the simultaneous occurrence of different marks at the single transcript resolution.

We thank the Reviewer for the suggestion. TandemMod is capable of identifying multiple modifications within the same read. Unlike NGS sequencing, Nanopore sequencing is a single-molecule sequencing method that can sequence entire transcripts. After TandemMod's feature extraction, we retain the 'read_id' for each single long transcript. The predicted modification sites with the same 'read_id' are associated with the same read. As suggested by the Reviewer, in this version, we have explored the simultaneous occurrence of m⁶A and m⁵C at the single transcript resolution. Firstly, we identified 4597 mRNAs with high-confidence m⁶A sites (modification rate >0.5) and 3945 mRNAs with high-confidence m⁵C sites (modification rate >0.5). Among these mRNAs, 2394 possess both m⁶A sites and m⁵C sites (**New Extended Data Fig. 15k**). Furthermore, we investigated the frequency of co-occurrence of m⁶A and m⁵C at the same long transcript for each gene. Genes that containing 30% of mRNA transcripts with occurrence of at least one m⁶A and at least one m⁵C were most abundant (**New Extended Data Fig. 15l**). For instance, we found that 58 out of 184 mRNA transcripts from *LOC_Os03g52840.1* gene contained both one m⁶A and one m⁵C modification, while 7 out of 41 mRNA transcripts from *LOC_Os03g20700.1* genes exhibited m⁶A and m⁵C co-occurrence (**New Fig. 6h-i**). We have included this analysis in our revised manuscript.

New Extended Data Fig. 15 k-l: **k**, Venn diagram showing overlap between genes containing high confident m⁶A-modified sites and genes with m⁵C-modified sites in rice WT sample. Genes containing sites with a modification rate greater than 0.5 are considered as high-confidence. Among these genes, 2394 contain both high confident m⁶A sites and m⁵C sites. **l**, The proportion of reads with simultaneous occurrence of m⁶A and m⁵C on the same read among the 2394 transcripts.

New Figure 6h-i, The statistics of reads with simultaneous occurrence of m⁶A and m⁵C from transcripts *LOC_Os03g52840.1* (h) and *LOC_Os03g20700.1* (i) in the rice WT sample.

Moreover, the Authors claim that detecting multiple RNA modifications simultaneously with Illumina-based methods is unfeasible (Line 47), but simultaneous profiling of m⁵C, m⁶A, and m¹A has been achieved using RBS-Seq (refer to <https://doi.org/10.1073/pnas.1817334116>). I recommend acknowledging this study and comparing TandemMod's results in HeLa cells with those obtained using RBS-Seq on the same cell line, as reported in the aforementioned paper.

We thank the Reviewer for providing this helpful reference. In this article (<https://doi.org/10.1073/pnas.1817334116>), RBS-Seq detected only 1944 m⁵C sites (of which 187 were labeled with high confidence) and no m¹A sites were detected in mRNA from HeLa cells by the proposed RBS-Seq method. We intersected the 1944 sites with all C sites detected by DRS data from WT HeLa cells, resulting in 1169 valid C sites. Among the 1169 C sites, 482 (41.2%) were reported by the TandemMod-m⁵C model with a predicted modification rate >0.2. We also intersected the aforementioned 187 high-confidence m⁵C sites with all C sites detected by DRS data from WT HeLa cells, resulting in 78 valid C sites, among which 40 (51.3%) were reported by the

TandemMod-m5C model with a predicted modification rate >0.2 . We acknowledged and cited the literature and revised the inappropriate expressions in our manuscript. Owing to the RBS-seq method identified much less modified sites than expected, we prefer not to include this analysis in our revised manuscript.

Additionally, the Authors perform an analysis on rRNAs, stating it sufficiently demonstrates that detecting one mark with TandemMod is not influenced by the presence of other RNA modifications. To strengthen this conclusion, I suggest running TandemMod, following its training on m6A, to detect m5C and PseudoU marks on IVET datasets to evaluate the impact of these modifications. The analysis could be repeated for all three RNA modifications. Moreover, while False Positive Rate (FPR) is crucial, assessing the impact of nearby RNA modifications on TandemMod's True Positive Rate (TPR) is equally essential to affirm that RNA modification proximity does not affect TandemMod.

We appreciate the constructive suggestion from the Reviewer. We have conducted an analysis to evaluate the performance of TandemMod across the IVET datasets with different types of modifications. To investigate whether nearby RNA modifications impact TandemMod's performance, we tested the TandemMod-m6A model on A sites from the IVET-m5C dataset and the TandemMod-m5C model on C sites from the IVET-m6A and IVET-m1A datasets (**New Extended Data Fig. 11a-c**). In the IVET-m5C dataset, we selected A sites with m5C distances ranging from -10 to 10 and evaluated the impact of the distance of neighboring modified sites on false positives of TandemMod. The results showed that the performance of TandemMod-m6A model was not significantly affected by the upstream or downstream m⁵C sites (from -10 to 10 nt), except for the closest C (+1 nt) with slightly increased FPR; Similarly, the performance of TandemMod-m⁵C model on C sites was not attenuated by the upstream or downstream m⁶A/m¹A sites (from -10 to 10) (**New Extended Data Fig. 12**). We also evaluated the feature importance, examining the attention of TandemMod paid to the five input bases. The results clearly demonstrate that the TandemMod model focused more on the centered base than the neighboring bases (**New Extended Data Fig. 12d**). This further illustrates that TandemMod tends to be immune to the impact from neighboring modified bases.

New Extended Data Fig. 12 Impact of other modifications on the identification of m⁶A and m⁵C in IVET datasets. **a**, Test the TandemMod-m⁶A model on A sites from the IVET-m⁵C dataset. We selected reads at different distances (-10 to 10) from m⁵C, predicted the modification probabilities of these A sites, and found that false positives were minimally influenced by neighboring modified sites. **b**, Test the TandemMod-m⁵C model on C sites from the IVET-m¹A dataset. **c**, Test the TandemMod-m⁵C model on C sites from the IVET-m⁶A dataset. **d**, The feature importance of the five consecutive bases learned by TandemMod. The results showed that although TandemMod takes 5-mer motif features as input, it focuses more on the central bases than on the flanking bases. This characteristic accounts for that TandemMod is not significantly influenced by neighboring modified bases.

2) The Authors present TandemMod as a tool suitable to profile RNA modifications at single base and single read resolution, i.e. a modification probability for sequenced base, and they support their claim based on the analysis of synthetic RNAs. The tool achieves median FP and TP rates of ~5% and ~90% respectively for m⁵C (Figure2G) and comparable FPR for other RNA modifications (Figure4D). These results are promising, nevertheless, I am not convinced they are good enough to claim such a fine resolution. I urge the Authors to extensively discuss the Precision of a classifier with a 5% FPR when applied to a dataset characterized by a significantly larger number of unmodified

sites compared to modified ones (Line 301). They should demonstrate that TandemMod's FPRs for claimed RNA modifications ensure sufficient precision based on the expected stoichiometries of the marks. A relevant yet indirect observation in this regard, it is surprising to me that m6A (the most abundant internal methylation in mRNAs) and m5C have the same "Read level modification rate" distribution (Figure 5A-D), which is also perfectly in line with the one presented in Figure 2G for the 0-stoichiometry condition (only driven by false positives). Despite the single base and single read resolution, TandemMod also provides a functionality to merge the information of reads spanning the same genomic location. As the Authors mention, this is extremely important to control false positives as clearly emerges comparing Figure 5B-E against Figure 5A-D. Due to the relevance of this point, I would move the description of this procedure from the methods to the results session, and I would include the analysis of TandemMod's performance at site resolution, with both balanced and unbalanced datasets, and an exhaustive discussion of parameters tuning as the Authors did for the read-level probability threshold.

We thank the Reviewer for raising this concern. We agree that a classifier with 5% FPR will introduce bias when applied to unbalanced dataset. The 0-stoichiometry condition in Figure 2g is indeed driven by false positives. To systematically investigate the impact of FPR on datasets with different proportions of modified reads, we generated mixed samples with modification ratio ranging from 0 to 1 with a step of 0.05. The site-wise modification rates predicted by TandemMod showed a gradually increase in these samples, in consistent with the ground truth (**New Extended Data Fig. 8a**). We then evaluated the error of modification rate between the predicted modification level and ground truth (**New Fig. 2f**). When predicting samples with a modification level of 0, the FPR was ~ 0.05 , and when predicting high-level modification sites, TandemMod exhibited a false negative rate (FNR) of ~ 0.1 . However, when predicting sites with modification levels ranging from 0.05 to 0.3, the predicted modification level closely aligns with the ground truth. This outcome is attributed to the offsetting effect of false positives and false negatives to some extent. We further explored TandemMod's performance at site resolution with both balanced and unbalanced datasets, evaluating the impact of site-level cutoff values on the true positive rate (TPR) (**New Extended Data Fig. 8b-d**). The TPRs dropped significantly when the site-level cutoff approached the ground truth modification level across all three mixed samples. This indicates that to achieve a high TPR, the site-level cutoff value should be lower than the true modification rate.

As suggested by the Reviewer, in this version, we have extensively discussed the potential shortcoming of TandemMod when applied to a dataset characterized by a significantly larger number of unmodified sites compared to modified ones at reads level, which other classifiers would also encounter. In the revised version, we have added the description of aggregation of read-level predictions to site-level predictions to the results section and discussed the impact of the modification rate for controlling false positive rate at site level.

New Extended Data Fig. 8: Site-level performance evolution of TandemMod on mixed m5C samples. **a**, The predicted site-level modification rates gradually increase align well with the ground truth stoichiometry increasing. Predicted site-level modification rates under various stoichiometries. The datasets used in this analysis were mixed from the ELIGOS-m⁵C dataset and the ELIGOS-normal C datasets. The distribution shows a gradual increase in predicted modification rates, aligning closely with the ground truth stoichiometry. **b**, The influence of site-level cutoff threshold on the true positive rate in the sample with m5C/C ratio of 0.1. **c**, The influence of site-level cutoff threshold on the true positive rate in the sample with m5C/C ratio of 0.3. **d**, The influence of site-level cutoff threshold on the true positive rate in the sample with m5C/C ratio of 0.5.

New Fig. 2f: TandemMod model and performance evaluation on published datasets. **f**, Boxplot showing the error between the predicted level and the ground truth level under different stoichiometries. The datasets used in this analysis were mixed from the ELIGOS-m⁵C dataset and the ELIGOS-normal C dataset. The TandemMod m⁵C model achieved low error when predicting sites with modification levels ranging from 0.05 to 0.3.

3) The Authors comment about the ambiguity of antibody-based methods (Line 66) but their claim is not clear to me. This is extremely important since, based on this sentence, they decided to avoid a comparison with Illumina based

methodologies to check the reliability of TandemMod predictions (except for a limited comparison with the rather old m6A-seq technique) relying almost exclusively on synthetic RNAs. Synthetic datasets are very informative but they lack important confounding factors: real sequence complexity, RNA modifications stoichiometry, gene expression levels heterogeneity, and other RNA marks. While several of these aspects have been partially addressed by the Authors (see also previous points), I strongly suggest extending the characterization of TandemMod's performance to real biological datasets also profiled with state of the art Illumina based methods. A great asset in this regard is the work recently published on Nature Communications by Zhong et al (<https://doi.org/10.1038/s41467-023-37596-5>) where 10 tools for m6A profiling were benchmarked against each other's and recent orthogonal Illumina techniques. Leveraging on the data and analyses presented by Zhong et al, the Authors could deeply characterize the performance of TandemMod in presence of all the relevant confounding factors previously mentioned. In my opinion, this analysis is mandatory also to support the claim that TandemMod is superior to existing tools for RNA modifications detection which is so far limited to few other methods compared in the context of synthetic RNAs (Line 226). Finally, the reported improvement in performance due to the sequence complexity retained by IVET compared to Eligos or Curlcake IVTs is minor, however, this could emerge more strikingly when estimating performances based on real biological datasets and Illumina orthogonal techniques.

We thank the Reviewer for these helpful suggestions. In line 66 we were intended to convey that the modification information deduced from Illumina based methodologies are not quite accurate to serve as training labels. However, we agree that the Illumina based methodologies can be used to evaluate the performance of the modification identification models in vivo. In the revised version, we compared the performance of TandemMod with nanom6A and m6Anet on the detection of m6A sites. As Zhong et al' research did not provide raw fast5 files, we used human K562 WT cell line to perform the evaluation. As nanom6A was designed for RRACH motifs and m6Anet was designed for DRACH motifs, we extracted RRACH motifs and DRACH motifs from the K562 DRS data, and compared them with nanom6A and m6Anet, respectively. The peaks from MeRIP-seq of K562 cell line were used as ground truth to evaluate the model performance. ROC-AUC and PR-AUC were compared between these methods. TandemMod achieved a ROC-AUC of 0.69 on RRACH motifs and a ROC-AUC of 0.67 on DRACH motifs, outperforms both nanom6A and m6anet (**New Fig. 5m-n**).

New Fig. 5: Validation of TandemMod on various human cell lines. **m**, Performance comparison of TandemMod and nanom6A on K562 RRACH motifs. Peaks from MeRIP-seq were used as ground truth. **n**, Performance comparison of TandemMod and m6Anet on K562 DRACH motifs. Peaks from MeRIP-seq were used as ground truth.

Furthermore, to access whether TandemMod model trained on the IVET datasets can perform better than the model trained on the curlcake dataset within biological sample. We compared the top 100, 500, 1000, 2000, 5000 and 10000 predicted m6A-modified sites from models trained on IVET and curlcake datasets. The analysis revealed that the top m6A-modified sites predicted by TandomMod trained from IVET libraries covered higher proportion of m6A-seq-validated sites compared to those predicted by TandomMod trained from Curlcake (**New Fig. 6b**). This finding suggested that the increased sequence complexity in IVET further improve the performance of TandemMod in detecting modifications within biological samples.

New Fig. 6: Transcriptome-wide profiling multiple types of rice RNA modifications under normal and high salinity environment. b, Bar plot showing the proportion of ranked m⁶A sites with high confidence validated by m⁶A-seq.

4) The Authors claim that TandemMod's design allows to leverage on a version of the algorithm trained on a specific RNA modification to learn how to classify a different mark more easily in terms of amount of required data and computational resource. However, I believe this session of the results is too qualitative. I recommend conducting a systematic analysis to measure classification performance, required training data, and computational resources for a standard instance of TandemMod designed for m⁶A detection compared to a counterpart trained using top layers trained on m⁵C. The Authors should quantitatively discuss how much TandemMod's design facilitates this purpose.

We appreciate the constructive feedback from the Reviewer regarding the adaptability and efficiency of the TandemMod algorithm. In response to the Reviewer's suggestion, we conducted a comprehensive analysis focusing on three key areas: classification performance, required training data, and computational resource utilization. This analysis compares a standard instance of TandemMod designed for m⁶A detection with its counterpart where the basic layers were initially trained on m⁵C.

Firstly, we recorded the training time for individual batches under two scenarios: transfer learning with pretrained TandemMod models and training from scratch. The results illustrate a significantly reduced training time per batch in the case of transfer learning compared to de novo training (New Fig. 4e). We also evaluated the efficiency regarding the amount of training data required when transferring a pretrained model to a new modification type versus starting the training from scratch. Our findings demonstrate that transferring a pretrained model demands considerably less training data without compromising performance (New Fig. 4f-g). Furthermore, we assessed the performance of the model across varying training epochs. This evaluation helped us understand the rate at which the model learns new modifications under both transfer learning and de novo training scenarios. The ROC and PR curves indicate that transfer learning not only reduces the computational resources and data requirements but also maintains a high standard of accuracy and efficiency (New Fig. 4e-f)

We hope this quantitative analysis satisfactorily addresses the Reviewer's concerns and illustrates the practical benefits of TandemMod's adaptable design.

Fig. 4: Multiple types of RNA modifications identified by TandemMod. **e**, Comparison of the running times between the transfer learning mode and training from scratch mode of TandemMod. The running time of transfer learning increases gradually with the increase in batch size, as compared to training from scratch. **f**, The ROC-AUC performance of the transfer learning mode and training from scratch mode of TandemMod was evaluated. In the transfer learning process, a pretrained m6A model was retrained on IVET datasets to obtain a m5C model. Meanwhile, the training from scratch process involved training a m5C model directly on the IVET dataset. It was observed that the transfer learning approach achieved higher performance levels more rapidly compared to training from scratch. **g**, The ROC-AUC performance comparison between the transfer learning mode and training from scratch mode of TandemMod using different sizes of training data. It is observed that the transfer learning approach achieves higher AUC with less training data.

Minor Comments:

5) Line 94: It would be interesting to extend analyses to all available RNA modifications from the dataset released by Jenjaroenpun et al., especially considering the significance of Inosine in the field.

We thank the Reviewer for this insightful suggestion. In response, we have expanded our base-level feature statistics to include 5-methyluridine (5moU) and Inosine (**New Extended Data Fig. 2a-d**). Consistent with our findings for other modifications, both 5moU and Inosine cause a significant decrease in base quality, accompanied by fluctuations in other relevant features. The results underscored the potential of using nanopore sequencing to identify 5moU and Inosine. Notably, Inosine, being a modified adenine base, is paired with C instead of T. Consequently, the Inosine-modified IVT sample comprises A, C, Inosine and U. In light of this, we conducted a comparative analysis of base-level features between Inosine and G, acknowledging the distinct pairing nature of Inosine with C. The significance of Inosine differs from that of the control sample in Jenjaroenpun et al.'s work because it is also compared to G.

New Extended Data Fig. 2 DRS features at base level and current level of 5moU and Inosine from the ELIGOS datasets. a, Base-level features of 5moU compared to normal U base. **b,** Base-level features of Inosine compared to G. **c,** Umap visualization of resampled current signals of 5moU and U under the sequence context CAUCA. **d,** Umap visualization of resampled current signals of Inosine and G under the sequence context CAUCA.

6) Line 119: Could the Authors expand on their considerations regarding the relative differences in current intensity between modified bases and their unmodified counterparts or other canonical nucleotides?

We thank the Reviewer for this suggestion. In Figure 1, we have demonstrated the differences in current between modified bases and their unmodified counterparts. In Extended Data Figure 1d, our aim was to show that modified bases can also be distinguished from other canonical bases. The PCA plots reveal that modified bases and other canonical bases form distinct clusters, thereby demonstrating their differences. To further illustrate this point, we have quantified the differences in the current signals between modified bases and other canonical bases (**New Extended Data Fig. 1e**).

New Extended Data Fig. 1: DRS features at base level and current level affected by 5-mer sequence contexts. e, The comparison of mean feature between the modified bases and the four canonical bases.

7) Figure 2B: The PR curves should include the expected performance of the null model, represented by a horizontal line with precision equal to the positive rate. Additionally, it would be useful to graphically depict the expected precision and recall for the default analysis setup.

We thank the Reviewer for this suggestion. We have added the expected performance of the null model for those PR curves where the y-axis begins below 0.5.

8) Figure 2E/Line 179: Does the filter on probabilities uniformly affect all 5-mers? It could be informative to present a plot similar to Figure 2E for each sequence context. Moreover, are the discarded instances enriched for specific sites?

We thank the Reviewer for raising these questions. Based on the Reviewer's suggestion, we have analyzed the distribution of predicted probabilities under different motifs. There are a total of 256 motifs with ± 2 bases neighboring the targeted base. For illustration purposes, we randomly selected 36 out of the 256 unique motifs using Python's random.sample function with a random seed of 0 (**New Extended Data Fig. 6-7**). In general, the probabilities for unmodified motifs are generally close to 0 and the probabilities for modified motifs are typically close to 1. Despite the general trend, we have noticed variations within these groups. For example, AACGG showed high confidence in both unmodified and modified states, however, AGCTT showed less confidence in its unmodified state. We applied a probability cutoff strategy to the prediction results, setting a threshold of 0.3/0.7, and analyzed the proportion of discarded reads for each motif (**New Extended Data Fig. 5**). The distribution of discarded reads was found to be uniform across all motifs under consideration.

New Extended Data Fig. 6: The predicted modification probabilities distribution for ELIGOS normal C sites. The 36 motifs were randomly selected from the 256 motifs using seed 0 with the random package in Python.

New Extended Data Fig. 7: The predicted modification probabilities distribution for ELIGOS m5C sites. The 36 motifs were randomly selected from the 256 motifs using seed 0 with the random package in Python.

New Extended Data Fig. 5: The proportion of discarded reads for the 256 motifs when applying a probability cutoff strategy with the threshold of 0.3-0.7. The distribution of discarded reads was found to be uniform across all motifs.

9) Line 211: Did the Authors test xPore on the 0-stoichiometry condition using two independent sets of unmodified reads?

We thank the Reviewer for raising this question. In Fig. 2g, the ELIGOS_normalC and ELIGOS_m5C datasets were used to generate mixtures with different m5C level. For the test of xPore on the 0-stoichiometry, the two sets of reads were all from the ELIGOS_normalC sample. We sampled different reads from the ELIGOS_normalC sample to do the comparison. Thanks for the comments, we have clarified this issue more clearly in the revised manuscript.

10) Lines 254-258: How does subsampling affect the number of genes represented in the training dataset and, consequently, the represented sequence complexity?

We thank the Reviewer for raising these questions. We trained the TandemMod models using the 2,473 genes that are common across the four IVET datasets. Although the train-test split does not change the number of genes in the training dataset, it does affect the diversity of sequences. We analyzed the sequence complexity of different motif lengths in the training set and found that the training set consistently exhibits a good level of complexity (**New Extended Data Fig. 9e**).

New Extended Data Fig. 9: Statistics of direct RNA sequencing results from the IVET datasets. e, The sequence coverage of the four IVET training data.

11) Line 272: The concept of “cross-device” applicability requires clarification. Does this conclusion depend on the sequencing platform used to collect the different datasets (e.g., MinION vs. GridION)?

We thank the Reviewer for pointing this out. The IVET datasets were produced using the ONT GridION platform, while the Curlcake datasets were generated using both GridION and MinION platforms. Additionally, the ELIGOS datasets were exclusively generated on the MinION platform. These datasets originate from distinct equipment and sequencing platforms. In this paragraph, we were intended to illustrate that pretrained TandemMod models can be applied to different sequencing equipment and platforms. We have revised the text to make it clearer to the readers.

12) Extended Figure 6B: The color scale is missing.

We thank the Reviewer for pointing this out. We have added the color bar to in the previous Extended Figure 13.

13) Typos: Line 26: sties -> sites; Line 929: l -> bold “i”

We thank the Reviewer for pointing this out. We have corrected these typos.

Reviewer #2 (Remarks to the Author):

I co-reviewed this manuscript with one of the reviewers who provided the listed reports as part of the Nature Communications initiative to facilitate training in peer review and appropriate recognition for co-reviewers.

We thank the Reviewer again for his/her time of reviewing our work and have provided us these constructive suggestions.

Reviewer #3 (Remarks to the Author):

This manuscript by Wu et al authors proposed a robust deep learning framework for identification of multiple types of RNA modifications from direct RNA sequencing data. Their model was trained based on carefully designed dataset with sufficient sequence diversity, which greatly addressed the major limitation of IVT-based methods. They also performed systematic tests with both in vitro and in vivo transcribed dataset. The work is very timely. I believe there are multiple research groups working on essentially the same projects.

I have the following comments.

1. Although authors used cDNA library as template to get RNA molecules of diverse sequence diversity, the generated dataset may not perfectly simulate the real modification status. For example, in 5-mer “AAAAA”;, all the five adenosine will be completely replaced by modified bases into “m⁶A”;, how this issue is addressed?

We thank the Reviewer for raising this question. We agree that using a cDNA library for generating diverse RNA sequences has its limitations in perfectly simulating real RNA modification statuses. In our study, we have attempted to mitigate this by training one model on all 256 motifs (Unlike nanom6A that trains one model for each motif). The TandemMod model trained on diverse sequence contexts have improved generalization performance, enabling it to effectively handle motifs with either single or multiple modifications. In the TandemMod model, although a sequence of five consecutive bases is utilized as input, it's important to note that the influence of each base is not uniform. To investigate which base the model predominantly focuses on, we conducted an evaluation of base-level feature importance of the trained TandemMod model. This was achieved by iteratively altering the features corresponding to each of the five bases and comparing the performance against the unaltered, baseline results (**New Extended Data Fig. 12d**). The results clearly demonstrate that the TandemMod model focused more on the centered base than the neighboring bases. Additionally, we tested the false positive rates on yeast rRNA data (**New Extended Data Fig. 13**) which suggest that our prediction result is not influenced by the neighboring modified bases. Nevertheless, we acknowledge that completely replicating the modification status in vivo remains challenging and future

studies could explore using both in vivo and in vitro data for training modification identification models.

New Extended Data Fig. 12 Impact of other modifications on the identification of m⁶A and m⁵C in IVET datasets. d, The feature importance of the five consecutive bases learned by TandemMod. The results showed that although TandemMod takes 5-mer motif features as input, it focuses more on the central bases than on the flanking bases. This characteristic accounts for that TandemMod is not significantly influenced by neighboring modified bases.

2. Authors performed comprehensive comparisons with basic machine learning methods, but not state-of-art methods. ONT-based dictation of RNA modifications is a very hot topic. There are new tools published monthly. I guess it is probably impossible to compare with the best existing approaches published in the past year, please try to get them cited in the introduction section. For example, DENA [1] and Penguin [2]. I am sure there are many more high impact works in this topic that should be cited at least, if not compared.

We thank the Reviewer for providing these valuable references. We have cited DENA and Penguin as well as several recently published researches to the introduction section.

3. Related to point 1, validation on human cell lines seems a little weak, not sufficient or strong. Specifically,

a. The performance for m⁵C modification does not look good. According to figure 5d-5e, the predicted results for WT and NSUN2-KO are not significantly differed, the difference in site-level results is even smaller than read-level, which looks weird.

We thank the Reviewer for raising this question. There are many m⁵C methyltransferases in eukaryotes. NSUN2 is one of the NSUN methyltransferase family (NOP2, NSUN2, NSUN4, NSUN5, NSUN7, etc.), and when NSUN2 is knocked out, other methyltransferases can still perform m⁵C modification. Therefore, in the NSUN2-KO sample, the level of m⁵C will decrease somewhat, but it won't be very significant. Please refer to the results from Hai-Li Ma et al. and Tang et al.'s research^{1, 2}.

Hai-Li Ma et al., 2023, Molecular Cell Tang et al., 2020, Developmental Cell

From Fig. 5d-5e, we believe that difference in site-level results is basically the same as read-level results. To quantify the difference, we calculated the p-value of modification rate of WT and NSUN2-KO samples at both the site level and the read level using t-test. The difference in site-level results (p-value 5.40e-298) is slightly more significant than read-level (p-value 6.25e-143). We hope this response could address the Reviewer's concern.

b. Also, if authors want to demonstrate the utility of their model on biological samples, they should compare different tools. For example, Tombo and ELIGOS for non-canonical base detection, m6Anet and nanom6A for m6A detection, etc.

We thank the Reviewer for this insightful suggestion. We agree with the Reviewer that more validations on biological samples are needed. In the revised manuscript, we compared the predicted m6A sites from the HEK293T cell line with m6ACE-Seq³ and miCLIP⁴ data from the same cell line. We used the 12050 m6A sites identified by miCLIP as ground truth, and compared the overlap with TandemMod predictions, m6ACE-seq and randomly selected A sites. Out of the reported 12,050 m6A sites, TandemMod successfully identified 1,183. Notably, the overlap ratio between TandemMod predictions and miCLIP data was found to be comparatively high when compared to m6ACE-seq and randomly selected A sites from DRACH motifs (**New Fig. 5o**).

New Fig. 5: Validation of TandemMod on various human cell lines. o, Validation of TandemMod on human HEK293T cell line. Sites from miCLIP were used as ground truth. Random sites and sites from m6ACE-seq were utilized for comparison.

To further validate Tandem's performance on biological samples, we compared TandemMod with m6Anet and nanom6A using DRS data from the human K562 cell line. MeRIP-seq⁵ data were utilized as the ground truth, and the performance was

evaluated using ROC and PR curves. TandemMod achieved a ROC-AUC of 0.69 on RRACH motifs and a ROC-AUC of 0.67 on DRACH motifs, outperforming both nanom6A and m6Anet (**New Fig. 5m-n**).

New Fig. 5: Validation of TandemMod on various human cell lines. m, Performance comparison of TandemMod and nanom6A on K562 RRACH motifs. Peaks from MeRIP-seq were used as ground truth. **n**, Performance comparison of TandemMod and m6Anet on K562 DRACH motifs. Peaks from MeRIP-seq were used as ground truth.

4. Regarding signal resampling (Line 586), the 100 signal points are randomly sampled or follow a time sequence order? Please clarify.

It is our negligence that we did not clearly describe the signal resample procedure. The 100 signal points are sampled following a time sequence order. In nanopore sequencing, RNA/DNA molecules pass through the nanopore at varying speeds, while the sequencing equipment maintains a fixed sample frequency. Consequently, this leads to differing signal lengths for each base. To address this variability, we perform signal resampling to generate signals of uniform length, which are then used as input for our deep learning model. In mathematics, a spline is a special function defined piecewise by polynomials. Assume current signals for a nucleotide with a length of k data points $X = (x_1, x_2, \dots, x_k)$ were originally sampled from an interval $[a, b]$. Define a function S that maps the signals to \mathbb{R} ,

$$S: [a, b] \rightarrow \mathbb{R}$$

The interval $[a, b]$ can be further divided into $k-1$ ordered, disjoint subintervals according to each data point

$$[a, b] = [t_0, t_1) \cup [t_1, t_2) \cup \dots \cup (t_{k-2}, t_{k-1}]$$

For each subinterval, define a polynomial P_i that map the interval to \mathbb{R}

$$P_i: [t_i, t_{i+1}] \rightarrow \mathbb{R}$$

Then, the original current signals can be represented by piecewise function P_i

$$\begin{cases} S(t) = P_0(t), & t_0 \leq t < t_1 \\ S(t) = P_1(t), & t_1 \leq t < t_2 \\ \vdots & \vdots \\ S(t) = P_{k-2}(t), & t_{k-2} \leq t < t_{k-1} \end{cases}$$

To attain smoothness in these curves, we need to impose constraints that ensure adjacent intervals have the same n th-order derivatives. These smoothed curves are commonly referred to as splines:

$$\begin{cases} P_{i-1}^{(0)}(t) = P_i^{(0)}(t) \\ P_{i-1}^{(1)}(t) = P_i^{(1)}(t) \\ \vdots \\ P_{i-1}^{(n)}(t) = P_i^{(n)}(t) \end{cases}$$

By solving the above constraint equations, we can get the interpolation function $S(t)$. Then, equally divide the interval $[a, b]$ into subintervals $T' = [t'_0, t'_1, \dots, t'_{l-1}]$ with length l ($l = 100$ in this manuscript) and resample new current values X' from $S(t)$ at these l points.

$$X' = S(T')$$

In this script, we used 1-order spline interpolation to resample the current signals to fixed length. We have added the signal re-sampling details to the Methods section.

5. My biggest concern is about the main contribution of the manuscript. Performance wise, it is always difficult to claim it is the best, with new methods being published all the time. The main contribution of the work is its capability of predicting the most types of RNA modifications. Only six different types of RNA modifications were supported, which seems rather limited given that there are at least twelve different types of widely occurring RNA modification according to a previous work (m6A, m1A, m5C, m5U, m6Am, m7G, I, Am, Cm, Gm, and Um). Please consider supporting the prediction of A-to-I (or other modification type) in the model.

We appreciate the Reviewer's insightful comments regarding the scope of RNA modifications supported by our initial TandemMod model. In the manuscript, we have demonstrated TandemMod's transferability and versatility in RNA modification prediction. Our aim is to provide a robust and adaptable tool for researchers focusing on various RNA modifications. In response to the Reviewer's suggestion, we have expanded our model to include the prediction of additional RNA modification, A-to-I, which was previously not supported. We employed the ELIGOS Inosine dataset for

this purpose. The adaptation of our model to this dataset achieved a remarkable ROC-AUC of 0.97, indicating a high level of accuracy in prediction. These results are illustrated in **new Fig. 4d**. We conducted a validation of the adapted Inosine model using IVET normal A data (**Extended Data Fig. 13a**). The results from these tests showed a significantly low level of false positive rates, which underscores the reliability and precision of our model in identifying A-to-I modifications.

Unlike other modifications (m6A, m5C or pseudouridine), there exists unique challenges posed by A-to-I modification, where the I base pairs with C instead U and the I base is often sequenced as G, in NGS-based identification methods. To further enhance the utility of our model, we also developed an Inosine/G identification model. The ROC-AUC and the false positive validation for the Inosine/G model, as detailed in **Response Fig. 1**, demonstrate its exceptional capability to differentiate Inosine from Guanine accurately. The transferred Inosine/A and Inosine/G identification models have been updated to our GitHub repository. We hope that these enhancements and the additional models provided address the Reviewer's concerns satisfactorily. We are grateful for the opportunity to improve our work based on these insightful feedbacks.

New Fig. 4: Multiple types of RNA modifications identified by TandemMod. d, ROC curve and PR curve showing the performance evaluation of the Inosine/A model

Response Fig. 1: ROC curve and PR curve showing the performance evaluation of the the Inosine/G model

Reviewer #4 (Remarks to the Author):

TandemMod is a novel deep learning framework that allows identification of multiple RNA modifications in Nanopore sequencing data. TandemMod's ability to characterize RNA modifications at the read level and lack of requirement for control samples are significant improvements in Nanopore-based detection of modifications. This study represents an important step in moving to the next "phase" of epitranscriptomics research where scientists can investigate the effect of combination of RNA modifications on biology. To make TandemMod useful for the epitranscriptomics community, the authors developed several modes for TandemMod utilization. The authors present evidence necessary to support their conclusions, however, several points in the manuscripts needs to be addressed prior to publication. I am in favor of publication of this study if the raised issues are addressed by the authors.

We thank the Reviewer for all these positive comments. We have revised our manuscript addressing the comments and suggestions given by the Reviewer as below.

L.39: adding "transgenic" before "human" would make it more clear for the reader

We thank the Reviewer for pointing this out. We have added "transgenic" before "human" to make it clearer.

L.65 and 66: it is not clear which antibody-based immunoprecipitated experiments are being referred to here. Reference as well as a little further explanation is necessary.

We thank the Reviewer for pointing this out. Two of the methods, m6A-seq and miCLIP-seq, have been used as ground truth to train DRS-based models^{3, 4, 6}. We have added the corresponding references and rephased the text to make it clearer to the readers.

L.72: It is not clear why these specific modifications were chosen for IVET dataset. The researchers are investigating more than these three modification in the rest of the manuscript, so it is unclear here as well as later in the text why they are concentrating on these A-based modifications.

We thank the Reviewer for raising this question. While numerous RNA modifications exist, the majority of them are present at low levels. Notably, m⁶A, m⁵C, and m¹A emerge as the most prevalent modifications found in eukaryotic mRNAs. Among them, m⁶A is the most abundant and most-studied by researchers⁷. Thus, we focused on m⁶A, m⁵C and m¹A to build the IVET datasets. We have added the explanations to the revised manuscript.

L.75: Can TandemMod be used to Identify multiple modifications in the same read?

We thank the Reviewer for the suggestion. This question is also raised by Reviewer 1. It was our oversight not to clearly state this aspect. TandemMod is capable of identifying multiple modifications within the same read. After TandemMod's feature extraction, we retain the 'read_id' for each single long transcript. The predicted modification sites with the same 'read_id' are associated with the same read. As suggested by the Reviewer, in this version, we have explored the simultaneous occurrence of m6A and m5C at the single transcript resolution. Firstly, we identified 4597 mRNAs with high-confidence m⁶A sites (modification rate >0.5) and 3945 mRNAs with high-confidence m⁵C sites (modification rate >0.5). Among these mRNAs, 2394 possess both m⁶A sites and m⁵C sites (**New Extended Data Fig. 15k**). Furthermore, we investigated the frequency of co-occurrence of m6A and m5C at the same long transcript for each gene. Genes that containing 30% of mRNA transcripts with occurrence of at least one m⁶A and at least one m⁵C were most abundant (**New Extended Data Fig. 15l**). For instance, we found 58 out of 184 mRNA transcripts from *LOC_Os03g52840.1* gene contain both one m⁶A and one m⁵C modification, while 7 out of 41 mRNA transcripts from *LOC_Os03g20700.1* genes with m⁶A and m⁵C co-occurrence (**New Fig. 6h-i**). We have included this analysis in our revised manuscript.

New Extended Data Fig. 15 k-l: **k**, Venn diagram showing high confident m⁶A-modified and m⁵C-modified transcripts in rice WT sample. Transcripts containing sites with a modification rate greater than 0.5 are considered high-confidence. Among these transcripts, 2394 contain both m⁶A sites and m⁵C sites. **l**, The proportion of reads with simultaneous occurrence of m⁶A and m⁵C on the same read among the 2394 transcripts.

New Figure 6h-i, The statistics of reads with simultaneous occurrence of m⁶A and m⁵C from transcripts LOC_Os03g52840.1 (h) and LOC_Os03g20700.1 (i) in the rice WT sample.

L.92: Are the in vitro-transcribed datasets; ELIGOS datasets? If yes, this needs to be clearly indicated for the reader. As ELIGOS are described in detail later in the text, this appears as a different dataset.

We thank the Reviewer for pointing this out. Yes, the in vitro-transcribed datasets referred to here are the ELIGOS datasets. We have modified the description of ELIGOS datasets to ensure clarity and avoid any confusion for the reader. This modification should help in understanding the continuity and relevance of the ELIGOS datasets within the context of our study.

L. 106: In this paragraph the authors are describing how different parameters can be used to differentiate the 6 RNA modifications in the study. The data is presented in the figure, however, for the reader, it might be beneficial to also present this in a table where it is indicated exactly which parameter combination allows differentiation of the 6 modifications.

We thank the Reviewer for raising this discussion. It is our negligence to mislead the Reviewers. In this paragraph, we analyzed the difference of individual feature between unmodified base and modified base for several motifs as examples. These features are modification-specific and motif-specific. The combination of these features could not allow differentiation of modification universally for all of the 256 5-mer motifs. Instead, we used these 6 base-level features combining 500 current-level features as input to train TandemMod for differentiation of modifications from unmodified ones. We have revised this paragraph to make it clearer.

L.108: Could the authors explain what they mean by re-squiggle; as this might not be the term that the audience of Nature Communications is widely familiar with?

We thank the Reviewer for pointing this out. In nanopore sequencing, the electric current signal level data produced from a nanopore read is referred to as a squiggle. Base calling this squiggle information may contain some errors compared to a reference sequence. The re-squiggle algorithm defines a new assignment from squiggle to reference sequence, hence a re-squiggle. We have revised the relevant text to ensure that readers can easily comprehend the data processing steps.

L.110-111: Could the authors explain what they mean by signal re-sampling with spline interpolation?

In nanopore sequencing, RNA/DNA molecules pass through the nanopore at varying speeds, while the sequencing equipment maintains a fixed sample frequency. Consequently, this leads to differing signal lengths for each base. To address this variability, we perform signal resampling to generate signals of uniform length, which are then used as input for our deep learning model. In mathematics, a spline is a

special function defined piecewise by polynomials. Assume current signals for a nucleotide with a length of k data points $X = (x_1, x_2, \dots, x_k)$ were originally sampled from an interval $[a, b]$. Define a function S that maps the signals to \mathbb{R} ,

$$S: [a, b] \rightarrow \mathbb{R}$$

The interval $[a, b]$ can be further divided into $k-1$ ordered, disjoint subintervals according to each data point

$$[a, b] = [t_0, t_1) \cup [t_1, t_2) \cup \dots \cup (t_{k-2}, t_{k-1}]$$

For each subinterval, define a polynomial P_i that map the interval to \mathbb{R}

$$P_i: [t_i, t_{i+1}] \rightarrow \mathbb{R}$$

Then, the original current signals can be represented by piecewise function P_i

$$\begin{cases} S(t) = P_0(t), & t_0 \leq t < t_1 \\ S(t) = P_1(t), & t_1 \leq t < t_2 \\ \vdots & \\ S(t) = P_{k-2}(t), & t_{k-2} \leq t < t_{k-1} \end{cases}$$

To attain smoothness in these curves, we need to impose constraints that ensure adjacent intervals have the same n th-order derivatives. These smoothed curves are commonly referred to as splines:

$$\begin{cases} P_{i-1}^{(0)}(t) = P_i^{(0)}(t) \\ P_{i-1}^{(1)}(t) = P_i^{(1)}(t) \\ \vdots \\ P_{i-1}^{(n)}(t) = P_i^{(n)}(t) \end{cases}$$

By solving the above constraint equations, we can get the interpolation function $S(t)$. Then, equally divide the interval $[a, b]$ into subintervals $T' = [t'_0, t'_1, \dots, t'_{l-1}]$ with length l and resample new current values X' from $S(t)$ at these l points.

$$X' = S(T')$$

In this script, we used 1-order spline interpolation to resample the current signals to fixed length. We have added the signal re-sampling details to the Methods section.

L. 116: Could the authors explain what 2D Umap space shows/indicates? Additionally, how does the specific sequence around the modified nucleotide contribute to clustering?

We acknowledge our oversight in not clearly defining the term 'UMAP'. UMAP, which stands for Uniform Manifold Approximation and Projection, is a manifold learning technique utilized for dimensionality reduction. The resampled signals with 500 features corresponding to each 5-mer sequence exist in a high-dimensional space. Our objective is to visualize these signals in a low-dimensional space. We have

applied UMAP transformation to the resampled signals, converting them into 2-dimensional data. This allows for their visualization in a scatter plot. We have added more details as well as the reference to the revised manuscript to make it more clear to the readers. [line No.] As demonstrated in Extended Figure 1a, the sequence surrounding the modified nucleotide significantly influences the current signals. To effectively compare the current differences between the modified base and its unmodified counterpart, it is essential that the flanking bases remain identical. In Figure 1d, the sequence motifs are AGCCA, UGAGU, ACUAA, and UUGUA for the four respective subplots. We have included these sequence motifs in the figure legend for clarity.

L.202-226: For the experiments described in these paragraphs, it is not clear why these specific modifications (m⁵C and m⁶A) were selected for each of these? It is not clear whether the same result would be expected for the other modifications mentioned in the manuscript.

We thank the Reviewer for raising this question. m⁶A and m⁵C are the most prevalent modifications in eukaryotic mRNAs, and a bunches of ML models for specifically detecting only m⁶A or m⁵C have been developed. Therefore, we firstly investigated TandemMod's performance on detecting m⁵C and m⁶A comparing several representative available tools. Subsequently, we utilized transfer learning to detect other modifications like m⁷G, I, and Ψ (**New Fig. 4a-d**). In this revised version, we further performed a systematic analysis, demonstrating that transferring a pretrained model demands considerably less training data without compromising performance (**New Fig. 4e-g**). We have added this in the result.

New Fig. 4: Multiple types of RNA modifications identified by TandemMod. **a-d**, ROC curve and PR curve showing the performance evaluation of m⁷G model (**a**), hm⁵C model (**b**), Ψ model (**c**) and Inosine model (**d**) transferred from m⁵C model and tested on ELIGOS datasets. **e**, Comparison of the running times between the transfer learning mode and training from scratch mode of TandemMod. The running time of transfer learning increases gradually with the increase in batch size, as compared to training from scratch. **f**, The ROC-AUC performance of the transfer learning mode and training from scratch mode of TandemMod was evaluated. In the transfer learning process, a pretrained m⁶A model was retrained on IVET datasets to obtain a m⁵C model. Meanwhile, the training from scratch process involved training a m⁵C model directly on the IVET dataset. It was observed that the transfer learning approach achieved higher performance levels more rapidly compared to training from scratch. **g**, The ROC-AUC performance comparison between the transfer learning mode and training from scratch mode of TandemMod using different sizes of training data. It is observed that the transfer learning approach achieves higher AUC with less training data.

L. 236: Construction of IVET database (Figure 3a) needs to be described in more details in the methods section. For example, in L. 533 it is not clear how the

cDNA pGADT7 library was constructed. Is it a commercially available library? If so, please refer to where it can be obtained. If the researchers generated it themselves, methods need to be updated with this information. For the experiments generating IVET training sets, how would incomplete replacement of ATP (i.e., when let's say 50% of ATP is unmodified) affect training of the model and identification of modification? Does complete replacement of unmodified nucleotide with modified nucleotide limit the diversity of sequences that are used for training of the data?

We thank the Reviewer for raising this concern regarding the construction of IVET datasets. The cDNA library containing cDNA of rice seedlings and inflorescence in plasmid pGADT7, was commercially available from OE BioTech (Shanghai, China). TandemMod was designed for training with individual reads. When partially replacing unmodified ATP with m6ATP, it was difficult to determine which reads were modified or not. Given that the training set requires ground truth labels, the partially modified sample is unsuitable to serve as training data. The complete replacement of unmodified nucleotides with modified nucleotides does have its limitations, and this concern was also raised by the Reviewer 3. However, we have attempted to mitigate this by training one model on all 256 motifs (unlike nanom6A, which trains one model for each motif). The TandemMod model trained on diverse sequence contexts has improved generalization performance, enabling it to effectively handle motifs with either single or multiple modifications. In the TandemMod model, although a sequence of five consecutive bases is utilized as input, the influence of each base is not uniform. We conducted an evaluation of base-level feature importance of the trained TandemMod model and found that the TandemMod model focused more on the centered base than on the neighboring bases (**New Extended Data Fig. 12d**). Additionally, we tested the false positive rates on unmodified IVET dataset and yeast rRNA data (**New Extended Data Fig. 11, New Extended Data Fig. 12 and New Extended Data Fig. 13**), which suggest that our prediction result is not influenced by the neighboring modified bases.

We acknowledge that the IVET datasets may not perfectly simulate real-world RNA modification statuses. However, they still remain valuable resources for training modification detection models.

L. 272: It is not clear what the authors mean by 'cross-device' applicability.

The IVET datasets were produced using the ONT GridION platform, while the Curlcake datasets were generated using both GridION and MinION platforms. Additionally, the ELIGOS datasets were exclusively generated on the MinION platform. These datasets originate from distinct equipment and sequencing platforms. In this paragraph, we were intended to illustrate that pretrained TandemMod models can be applied to different sequencing equipment and platforms. We have revised the text to make it clearer to the readers.

L.336: Not all of the modifications clustered in distinct areas in Fig. 4f. Could the author comment on what consequences this lack of distinct clustering for some modifications has in terms of using TandemMod in the future?

We thank the Reviewer for raising this question. Figure 4f illustrates a scatter plot depicting the UMAP visualization of TandemMod-processed features. This two-dimensional representation serves the purpose of visualization. While some modifications appear close to each other in this 2-D space, they can be distinguished in higher-dimensional spaces. For instance, clusters of m⁵C and hm⁵C modifications appear similar in the UMAP plot. By leveraging base-levels and event features, TandemMod can be designed to differentiate between hm⁵C and m⁵C to some extent, although its performance is relatively lower compared to the TandemMod model for distinguishing C and m⁵C. This discrepancy is likely attributed to the high similarity between m⁵C and hm⁵C. With advancements in Nanopore technology, direct RNA sequencing is becoming increasingly accurate. And the performance of TandemMod in distinguishing similar modifications is expected to improve. We have added this comment to the discussion section in the revised manuscript.

L.365: Why were these m6A sites specifically reliable?

We thank the Reviewer for pointing this out. The word “reliable” is not properly used. We were intended to convey that some sites are known as m6A-modified according to previous research. We have corrected this sentence to “We then focused on site 1216 of the ACTB transcript and site 1339 of the BSG transcript, which were reported as known m6A sites.”

L.368: Why were top 30 gene selected? How do these identified genes compared to the previously published data on m6A-epitranscriptome? I think in order to demonstrate that the method is working in vivo, the authors need to perform a more thorough comparison to the available data. Identification of the DRACH motif should not be the only measure of reliability assessment. This point also applied to the m5C data in the next paragraph (L.380-390).

We thank the Reviewer for raising these questions. We selected the top 30 genes for the visualizing the identified differentially m6A-modified sites. We agree with the Reviewer that more validation on in vivo data using available data is needed. In the revised manuscript, we compared the predicted m6A sites from the HEK293T cell line with m6ACE-Seq³ and miCLIP⁴ data from the same cell line. We used the 12050 m6A sites identified by miCLIP as ground truth, and compared the overlap with TandemMod predictions, m6ACE-seq and randomly selected A sites. Out of the reported 12,050 m6A sites, TandemMod successfully identified 1,183 m6A sites. Notably, the overlap ratio between TandemMod predictions and miCLIP data was found to be comparatively high when compared to m6ACE-seq and randomly selected A sites from DRACH motifs (**New Fig. 5o**).

New Fig. 5: Validation of TandemMod on various human cell lines. o, Validation of TandemMod on human HEK293T cell line. Sites from miCLIP were used as ground truth. Random sites and sites from m6ACE-seq were utilized for comparison.

To further validate TandemMod's performance on biological samples, we compared TandemMod with m6Anet and nanom6A using DRS data from the human K562 cell line. MeRIP-seq⁵ data were utilized as the ground truth, and the performance was evaluated using ROC and PR curves. TandemMod achieved a ROC-AUC of 0.69 on RRACH motifs and a ROC-AUC of 0.67 on DRACH motifs, outperforming both nanom6A and m6Anet (**New Fig. 5m-n**).

New Fig. 5: Validation of TandemMod on various human cell lines. m, Performance comparison of TandemMod and nanom6A on K562 RRACH motifs. Peaks from MeRIP-seq were used as ground truth. **n**, Performance comparison of TandemMod and m6Anet on K562 DRACH motifs. Peaks from MeRIP-seq were used as ground truth.

L. 483: In the discussion pertaining to accuracy on prediction on short reads, could the authors comment on the subsets of biologically relevant RNAs that TandemMod might not be applicable to (for example, small non-coding RNAs)? Is there a way to modify TandemMod in the future to make it work on short reads?

We thank the reviewer for raising this question. In nanopore sequencing, there is a tendency for the initial and terminal segments of a sequence to have lower sequencing quality. If the read length is too short, there is a higher likelihood of encountering a proportion of low-quality 5-mers, which can result in a degradation of the model's performance. TandemMod is designed for detecting modifications on mRNAs. When applied to short reads, such as small non-coding RNAs and tRNA, TandemMod may encounter performance degradation. To apply it to short sequences, additional adapters can be attached to both sides of the short sequence to turn it into a longer sequence, and then DRS sequencing can be performed. We have included these comments in the discussion section.

References

1. Ma, H.L. et al. SRSF2 plays an unexpected role as reader of m(5)C on mRNA, linking epitranscriptomics to cancer. *Mol Cell* **83**, 4239-4254 e4210 (2023).
2. Tang, Y. et al. OsNSUN2-Mediated 5-Methylcytosine mRNA Modification Enhances Rice Adaptation to High Temperature. *Dev Cell* **53**, 272-286 e277 (2020).
3. Koh, C.W.Q., Goh, Y.T. & Goh, W.S.S. Atlas of quantitative single-base-resolution N(6)-methyl-adenine methylomes. *Nat Commun* **10**, 5636 (2019).
4. Linder, B. et al. Single-nucleotide-resolution mapping of m6A and m6Am throughout the transcriptome. *Nat Methods* **12**, 767-772 (2015).
5. Dou, X. et al. RBFOX2 recognizes N(6)-methyladenosine to suppress transcription and block myeloid leukaemia differentiation. *Nat Cell Biol* **25**, 1359-1368 (2023).
6. Hendra, C. et al. Detection of m6A from direct RNA sequencing using a multiple instance learning framework. *Nat Methods* **19**, 1590-1598 (2022).
7. Shi, H., Chai, P., Jia, R. & Fan, X. Novel insight into the regulatory roles of diverse RNA modifications: Re-defining the bridge between transcription and translation. *Mol Cancer* **19**, 78 (2020).

Reviewer #1 (Remarks to the Author):

I would like to thank the authors for their effort since they extensively addressed the majority of the points I raised.

There is one aspect of the work that I suggest to further explore, which concerns the applicability of TandemMod to profile RNA modifications at single base and single-read resolution. As mentioned in my initial review, I am concerned that a FPR of 5%, when applied to a dataset characterized by a significantly larger number of unmodified sites compared to modified ones, could compromise the tool's performance. Indeed, for highly unbalanced datasets, the fraction of TP hits could be very low, hampering the usefulness of TandemMod's predictions. If the majority of the true hits a user gets are, in fact, false, the analysis becomes poorly informative. To this regard, I would recommend extending the analysis performed on unbalanced datasets (so far based on TPR, FPR, and TNR) by including a discussion on the classifier precision ($TP/(TP+FP)$).

Finally, regarding the precision-recall curves presented in Figure 5, I am not convinced by the null model. It suggests that the dataset is balanced and that the tools negatively select modified sites for "higher" recalls. Is this the case? This observation is not in agreement with published results, for example please see <https://doi.org/10.1093/bib/bbae001> Figure 3.

Reviewer #1 (Remarks on code availability):

The github web page releasing the tool source code seems well prepared. The tool documentation is useful and sufficiently detailed.

Even though I haven't tried installing and executing the code itself, the provided documentation seems fully adequate to support its installation and execution.

Furthermore, scripts to reproduce key findings reported in the manuscript figures were also released.

Reviewer #2 (Remarks to the Author):

I co-reviewed this manuscript with one of the reviewers who provided the listed reports as part of the Nature Communications initiative to facilitate training in peer review and appropriate recognition for co-reviewers.

Reviewer #3 (Remarks to the Author):

All my comments have been properly addressed. I am happy to recommend the manuscript to be accepted at NC.

Reviewer #4 (Remarks to the Author):

The manuscript reviewed by the authors have further validated their transferable deep learning approach for identification of multiple RNA modifications with Nanopore direct RNA sequencing. The authors have responded to my comments either by further explaining obtained results or presenting new experiments. Identification of multiple RNA modifications in a single sequencing read within a single experiment is an important step in the field of epitranscriptomics. I am therefore in favor of publishing this study.

REVIEWER COMMENTS

Reviewer #1 (Remarks to the Author):

I would like to thank the authors for their effort since they extensively addressed the majority of the points I raised.

There is one aspect of the work that I suggest to further explore, which concerns the applicability of TandemMod to profile RNA modifications at single base and single-read resolution. As mentioned in my initial review, I am concerned that a FPR of 5%, when applied to a dataset characterized by a significantly larger number of unmodified sites compared to modified ones, could compromise the tool's performance. Indeed, for highly unbalanced datasets, the fraction of TP hits could be very low, hampering the usefulness of TandemMod's predictions. If the majority of the true hits a user gets are, in fact, false, the analysis becomes poorly informative. To this regard, I would recommend extending the analysis performed on unbalanced datasets (so far based on TPR, FPR, and TNR) by including a discussion on the classifier precision ($TP/(TP+FP)$).

We thank the Reviewer for this constructive suggestion regarding the application of TandemMod in profiling RNA modifications at single-base and single-read resolution. In this revision, we further analyzed and discussed the precision of a model with an FPR of ~5% when applied to an unbalanced dataset. To achieve this, we generated 21 mixed samples with m5C ratios ranging from 0 to 1 with a step of 0.05 and tested TandemMod on these samples. We evaluated the classifier's precision ($TP/(TP+FP)$) for each site and observed a decrease in precision in samples with a low m5C rate, alongside a progressive improvement in precision as the modification rate increased (**New Extended Data Fig. 8e**). To explore whether the probability cutoff strategy adapted in this study could improve the model performance on unbalanced data, we adjusted the probability cutoff from the default 0.5-0.5 to 0.1-0.9 and found that the precision significantly improved on these samples (**New Extended Data Fig. 8f**). For the unbalanced sample with an m5C rate of 0.05, the mean precision improved from 0.37 to 0.65. Although the mean precision is considerably improved, there are still fluctuations ranging from 0.06 to 0.95. As the Reviewer expected, there will still be a proportion of false positives when applying the tool to the sites with low-modification rate.

Despite this challenge, TandemMod is valuable for identifying RNA modification sites with higher modification rate (>20%) at read level using 0.1-0.9 probability cutoff

(New Extended Data Fig. 8f). While TandemMod can offer modification probabilities for every single base within a single read, only those modification sites detected by TandemMod with high confidence (e.g. modification rate higher than 20% and reads more than 10) are recommended for further examination at read level (single-base and single-read resolution). We have added this in the result and discussion section.

New Extended Data Fig. 8: Site-level performance evolution of TandemMod on mixed m5C samples. **a**, The predicted site-level modification rates gradually increase align well with the ground truth stoichiometry increasing. Predicted site-level modification rates under various stoichiometries. The datasets used in this analysis were mixed from the ELIGOS-m⁵C dataset and the ELIGOS-normal C datasets. The distribution shows a gradual increase in predicted modification rates, aligning closely with the ground truth stoichiometry. **b**, The influence of site-level cutoff threshold on the true positive rate in the sample with m5C/C ratio of 0.1. **c**, The influence of site-level cutoff threshold on the true positive rate in the sample with m5C/C ratio of 0.3. **d**, The influence of site-level cutoff threshold on the true positive rate in the sample with m5C/C ratio of 0.5. **e**, The model precision on the mixed samples with probability cutoff of 0.5-0.5. The precision is low where the sample is unbalanced. **f**, The model precision on the mixed samples with probability cutoff of 0.1-0.9. The precision was significantly improved, indicating the effectiveness of applying a higher cutoff value.

Finally, regarding the precision-recall curves presented in Figure 5, I am not convinced by the null model. It suggests that the dataset is balanced and that the tools negatively select modified sites for "higher" recalls. Is this the case? This observation is not in agreement with published results, for example please see <https://doi.org/10.1093/bib/bbae001> Figure 3.

We thank the Reviewer for providing this informative reference and pointing this out. We made a mistake in the previous revision by misunderstanding about the "null" model. For the precision-recall curves in Figure 5, the dataset is DRS data from K562 cell line and the data is unbalanced. In this revision, we used a random classifier, which randomly generated a modification probability for each input, as the null model and updated the ROC and precision-recall curves (**New Figure 5m, n**). We sincerely thank the Reviewer once again for the careful review, which helps us a lot to improve our manuscript.

New Fig. 5m-n: Validation of TandemMod on various human cell lines. m, Performance comparison of TandemMod and nanom6A on K562 RRACH motifs. Peaks from MeRIP-seq were used as ground truth. A random classifier which randomly generated a modification probability for each input was used as null control for evaluate the model performance. **n,** Performance comparison of TandemMod and m6Anet on K562 DRACH motifs. Peaks from MeRIP-seq were used as ground truth.

Reviewer #1 (Remarks on code availability):

The github web page releasing the tool source code seems well prepared. The tool documentation is useful and sufficiently detailed. Even though I haven't tried installing and executing the code itself, the provided documentation seems fully adequate to support its installation and execution. Furthermore, scripts to reproduce key findings reported in the manuscript figures were also released.

We thank the Reviewer again for taking the time to review our manuscript.

Reviewer #2 (Remarks to the Author)

I co-reviewed this manuscript with one of the reviewers who provided the listed reports as part of the Nature Communications initiative to facilitate training in peer review and appropriate recognition for co-reviewers.

We thank the Reviewer again for his/her time of reviewing our work and have provided us these constructive suggestions.

Reviewer #3 (Remarks to the Author)

All my comments have been properly addressed. I am happy to recommend the manuscript to be accepted at NC.

We sincerely appreciate the insightful feedback provided by the Reviewer, which has significantly contributed to elevating the quality of our work.

Reviewer #4 (Remarks to the Author)

The manuscript reviewed by the authors have further validated their transferable deep learning approach for identification of multiple RNA modifications with Nanopore direct RNA sequencing. The authors have responded to my comments either by further explaining obtained results or presenting new experiments. Identification of multiple RNA modifications in a single sequencing read within a single experiment is an important step in the field of epitranscriptomics. I am therefore in favor of publishing this study.

We are grateful for the valuable feedback from the Reviewer, which has greatly enhanced the quality of our work.